# HealthLoopQA: A Context-Aware Question Answering Benchmark for Interpreting Wearable Monitoring Data in Diabetes Care

## Abstract

Medical wearables are transforming chronic disease management by enabling continuous physiological monitoring and personalised therapy, improving both clinical outcomes and quality of life. As these systems become integrated into daily care, interpreting long-term monitoring data is critical for patients and clinicians to understand health trends, detect safety-critical events promptly, and make informed decisions. However, this requires in-depth temporal reasoning that integrates domain knowledge, patient-specific conditions, and system-level behaviours, challenges that go beyond traditional time-series tasks. Recent advances in large language models (LLMs) offer new opportunities for context-aware reasoning and natural language interaction with medical monitoring data. Yet, existing question answering (QA) benchmarks lack the contextual richness, reasoning depth, and fault modelling required for realistic long-term medical monitoring scenarios. We introduce HealthLoopQA to bridge this gap. HealthLoopQA includes a hybrid closed-loop insulin delivery testbed that simulates realistic physiological and therapeutic monitoring data under varied patient activity schedules and 17 fault scenarios reflecting device failures and cybersecurity threats. The benchmark comprises comprehensive domain-specific QA templates for training and evaluating models, covering process mining, anomaly detection, and predictive reasoning, categorised by reasoning depth, ranging from purely descriptive statistics to causal and inferential reasoning. Each QA pair includes both a numerical answer and a textual rationale, enabling assessment of quantitative accuracy and reasoning fidelity. We evaluated prompt-based and agent-based baselines with state-of-the-art LLMs. Failure analysis reveals a broader phenomenon of *In-Context Laziness*, where the model substitutes full computations with rough approximations and confident narrative justifications, highlighting the limitations of current LLMs for structured long-term time-series reasoning. HealthLoopQA aims to facilitate the development of in-depth and trustworthy time-series understanding in AI systems for digital health.

## 1 Introduction

For millions of people with type 1 diabetes, Automatic Insulin Delivery (AID) systems—wearable physiological closed-loop control systems that continuously monitor blood glucose (BG) and automatically adjust insulin delivery to maintain BG in the safe range, representing the difference between intensive self-management with life-threatening complications and normal daily activities (Collyns et al., 2021; Renard, 2022; Godoi et al., 2023). Yet, with streams of BG measurements and life-critical insulin delivery decisions generated about every 5 minutes, challenges are posed for patients and clinicians to interpret them meaningfully and effectively (Mackett et al., 2023).

While recent medical AI assistant benchmarks have been introduced for patient inquiry dialogue (Arora et al., 2025), clinicians' daily workflows (Bedi et al., 2025), and specific medical monitoring tasks such as classification (Oh et al., 2023) or forecasting (Sergazinov et al., 2024), there remains no unified framework for evaluating medical wearable monitoring assistants (MWMA) that address the diverse patient monitoring needs of long-term continuous physiological measurements. Furthermore, existing conversational medical wearable monitoring benchmarks (Oh et al., 2023; Healey &

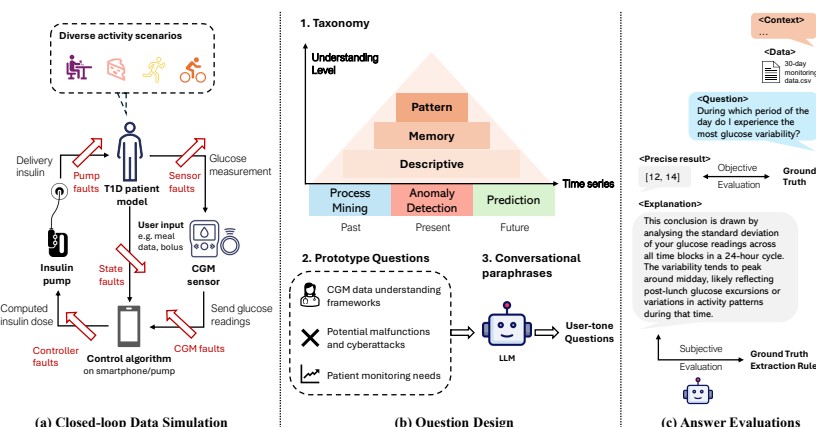

Figure 1: Overview of the proposed HealthLoopQA framework. (a) Closed-loop data simulation: simulates the closed loop between virtual T1D patients, CGM sensors, control algorithms, and insulin pumps, with fault modelling to reproduce possible malfunctions and interferences under diverse activity scenarios (e.g., meals, snacks, running, cycling). (b) Question design: a two-dimensional taxonomy is categorised by time-series task type and reasoning depth. Prototype questions are derived from CGM analysis frameworks, clinical guidelines, documented AID vulnerabilities, and patient monitoring needs. Based on these taxonomies and prototype questions, we generate 150 user-tone questions with LLMs. (c) Answer evaluation: manually defined reasoning instructions and automatic answer-extraction modules support evaluation based on numerical precision and reasoning alignment.

Kohane, 2024b) fail to capture essential contextual factors (e.g., therapy and patient activity) necessary for accurate physiological data interpretation, omit modeling of device failures (Kapadia, 2024) and cybersecurity threats (Niu & Lam, 2026) that impact real-world monitoring, and primarily focus on single-variate statistical analysis that oversimplifying the complex, dynamic patterns inherent in continuous physiological data.

To address these gaps, we introduce HealthLoopQA, an extensible evaluation benchmark for interpreting medical wearable monitoring data in diabetes care as illustrated in Figure 1. To ensure comprehensive and quantifiable evaluation, we adopt a QA format, which offers three key advantages (Gardner et al., 2019). First, QA naturally mirrors how patients inquire with a clinician about monitoring data, supporting easy and human-centred interaction. Second, it enables fine-grained assessment of diverse reasoning abilities—beyond traditional time-series tasks—covering numerical analysis, temporal reasoning, and causal inference, while accommodating textual information such as user input instructions and domain knowledge. Third, the QA format facilitates leveraging recent advances in large language models (LLMs) for zero/few-shot and multimodal reasoning (Reichman et al., 2025), while remaining compatible with modular hybrid systems that integrate time-series models with language agents.

Using the increasingly adopted AID system in diabetes care as an exemplar, where accurate interpretation of monitoring data is essential for improving long-term glycemic control (Millson & Hammond, 2020) and ensuring safe system operation in real time (Kapadia, 2024; Niu & Lam, 2026), we establish the MWMA task framework and corresponding performance standards. We further decompose the analytical process into key stages and atomic reasoning skills required for reliable interpretation, as illustrated in Figure 2 and detailed in the section 4. Our main contributions are as follows:

*(i)* We propose the **MWMA task framework** that formalises 3 analysis stages and 11 core reasoning abilities essential for establishing the foundation for systematic medical wearable data understanding. *(ii)* We develop a **closed-loop simulation testbed** supporting long-term simulation under diverse meal and exercise patterns and incorporates 17 fault patterns reflecting documented device malfunctions and cyber-physical threats encountered in real-world operation. *(iii)* We introduce **HealthLoopQA benchmark** with **150 patient-centred questions** grounded in daily diabetes self–monitoring needs. These questions span key time-series tasks, including **process mining, anomaly**

**detection, and prediction** and are organised by reasoning depth from descriptive to causal inference. *(iv)* We implement **automatically answer-extraction modules** that extract golden standard responses from raw monitoring data, along with reasoning rationale, enabling assessment of both **numerical accuracy and reasoning fidelity**. *(v)* We evaluate **2 prompt-based and 1 agent-based LLM baseline** based on GPT-5, revealing that out-of-the-box LLMs struggle with the required reasoning. We supplement quantitative evaluations with qualitative analysis that exposes specific pathologies in LLM behaviour, underscoring the benchmark's challenge and diagnostic value.

HealthLoopQA bridges the gap between traditional time-series analysis and the contextual understanding and diverse reasoning tasks required for real-world wearable monitoring. As medical wearable technologies proliferate, the need for AI systems that can reliably interpret continuous physiological data in context is becoming critical. HealthLoopQA offers both the infrastructure to measure progress and the analytical lens to evaluate the reasoning capabilities essential for trustworthy MWMA. Ultimately, it aims to accelerate the development of intelligent systems that enhance patient self-monitoring and clinical decision-making, improving outcomes for millions living with chronic conditions.

## 2 SCOPE

We constrain the task to one-turn natural language interactions that return objective or predictive answers to user queries about medical wearable monitoring data, without issuing diagnostic or behavioural instructions. The AID system testbed serves as the primary data source for enabling explicit modelling of faults in the loop, covering a cohort of virtual patients representative of the population of people with T1D, and systematic evaluation under controlled conditions. While the benchmark provides a systematic evaluation of key reasoning abilities, real-world deployment must account for additional complexities, as discussed in the Limitations, and undergo validation on real-world datasets and clinical studies. The proposed task framework and reasoning taxonomy generalise to other medical wearable applications (e.g., ECG, stress, sleep), but domain-specific datasets and questions must be tailored. While formulated as a QA task, the benchmark is not confined to language model evaluation; achieving the target reasoning standards likely requires hybrid systems that couple language models with time-series reasoning modules. The simulated dataset can also support training or assessing time-series tasks such as pattern recognition, anomaly detection, forecasting, and improvement of atomic reasoning abilities in the task framework.

## 3 RELATED WORK

**QA Benchmarking in Time Series Interpretation.** QA has been extensively studied in language and vision, but its application to time-series interpretation is relatively recent, enabled by advances in LLMs. Early work such as DeepSQA (Xing et al., 2021) relied on template-based questions with limited linguistic diversity and reasoning difficulty. TimeSeriesExam (Cai et al., 2024) introduced 5 core tasks with increasing difficulty to assess LLMs' general time-series reasoning capabilities, yet it does not involve domain-specific knowledge. SensorQA (Reichman et al., 2025) designed human-centred queries over long-duration data in daily-life monitoring, but mainly focused on process mining and evaluated responses using language-similarity metrics rather than numerical accuracy. MTBench (Chen et al., 2025) introduced regression and classification metrics for financial and weather data, yet omitted failure-aware monitoring in medical wearables.

In healthcare, Time-MQA (Kong et al., 2025) covered multiple domains including physiological data, but posed generic questions not tailored to medical wearables monitoring. Domain-specific efforts such as ECG-QA (Oh et al., 2023) and LLM-CGM (Healey & Kohane, 2024a) designed QA for electrocardiogram (ECG) and CGM monitoring, but remained limited to single modalities. These benchmarks also lacked therapeutic or activity context and did not model system failures. Overall, existing time-series QA benchmarks demonstrate potential for evaluating medical data interpretation but remain short of providing domain-specific, context-rich, and failure-aware evaluations.

**Understanding and Analysis of CGM Data.** Beyond QA benchmarking, substantial work has focused on CGM analysis and interpretation. Clinical guidelines such as the 2025 American Diabetes Association (ADA) Standards of Care emphasize CGM-derived metrics, highlighting the Ambulatory Glucose Profile (AGP) and Time in Range (TIR) as key measures for diabetes management

(ame, 2025). Statistical and machine learning approaches, including glucodensity curves and long short-term memory (LSTM) models, capture temporal trends and support clustering or forecasting, but often lack interpretability (Klonoff et al., 2025).

Recent reviews, e.g., CGM Data Analysis 2.0 (Klonoff et al., 2025), argue that traditional statistics oversimplify dynamic glucose fluctuations and highlight alternative frameworks including functional data analysis, AI/ML, and foundation models. These approaches enable richer interpretation of complex glucose patterns and support personalized decision-making. This underscores the need for QA pairs that move beyond traditional metrics to capture dynamic glucose trajectories and individualized treatment contexts.

More advanced architectures such as AttenGluco (Farahmand et al., 2025) integrate environmental data through cross-attention for long-term forecasting, while GLUCOBENCH (Sergazinov et al., 2024) introduces CGM-specific standards for prediction and uncertainty estimation. However, both remain narrow in scope.

**QA as Task vs. QA as Diagnostic Tool**. To establish LLMs' abilities to reason over CGM data, we rely on QA as a diagnostic tool (Srivastava et al., 2023), rather than performing costly human-centred experiments or evaluating task performance by offline proxy tasks (Bedi et al., 2025), that are mired with common NLG evaluation pitfalls(Gatt & Krahmer, 2018; Huang et al., 2021) and not always predict application performance (Doshi-Velez & Kim, 2017).Specifically, focus on the fundamental reasoning abilities that govern understanding, analysis, and inference, including physiological reasoning about glucose dynamics, temporal reasoning for pattern recognition in time-series data, and contextual integration of patient-specific factors. Natural language queries serve as the natural interface through which we probe these reasoning capabilities in LLMs. Thus we rely on QA as a task *format* (Gardner et al., 2019) rather than the task itself, where linguistic diversity and clinical plausibility of the queries would be more central.

## 4 ORGANIZATION OF BENCHMARK

### 4.1 TASK DEFINITION

We define the MWMA task as a system that responds to user inquiries about monitoring data with accurate answers based on continuous biosensor readings, therapy records and other relevant contextual information. The task simulates single-turn QA interactions between a patient and an MWMA. Given: $X$, the history of monitored physiological time-series data (e.g., 30 days of CGM readings); $C$, contextual information, such as insulin delivery records, patient profile, and activity logs; $Q$, a natural language question regarding the monitoring data; $I$, a reasoning instruction that describes the process or evidence gathering that leads to the final answer. The system $f$ is expected to output $A$, a precise answer, which can take one of several forms depending on the task: *(i)* a precise numerical value, *(ii)* a categorical class label, or *(iii)* a temporal attribute such as a timestamp or event duration. We therefore formalise the task as: $f : (X, C, Q, I) \rightarrow (A)$.

### 4.2 TASK FRAMWORK

We formalise the reasoning process in MWMA into three analytical stages encompassing eleven atomic reasoning abilities required for reliable interpretation, as illustrated in Figure 2 with example questions. This framework defines the desired cognitive and computational competencies of an intelligent assistant to deliver accurate and reliable insights. These atomic reasoning abilities are decomposed from the answer instruction we formed for each question template base on existing CGM data understanding frameworks and guidelines Klonoff et al. (2025); Bergenstal (2018); Millson & Hammond (2020). We put detailed definition, purpose, and reasoning examples of all atomic reasoning abilities in Appendix D with Table 2, and illustrate their organisation and short definitions here.

#### 4.2.1 INTENTION UNDERSTANDING

The first stage focuses on whether the assistant correctly understands the user's goal and instruction. It includes (1) **Question Understanding (QU)**: identifying user intention and the target task from questions (e.g., "What is my average BG this week?" vs. "Why did my BG go low last night?") (2) **Instruction Alignment (IA)**: ensuring reasoning follows the user's specified conditions or con-

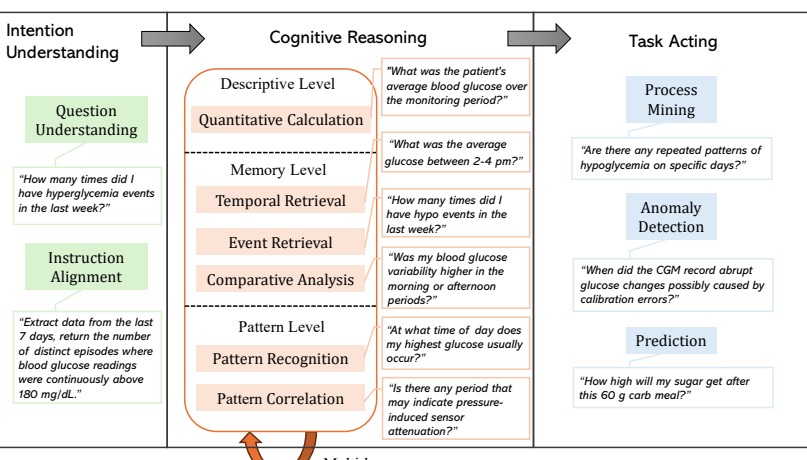

Figure 2: Task framework of the Medical Wearables Monitoring Assistant (MWMA). The MWMA pipeline comprises three stages. (1) Intention Understanding: interprets user queries and instructions. (2) Cognitive Reasoning: engages three levels of reasoning depth required by different monitoring goals: descriptive quantitative calculation; memory-based reasoning that supports temporal retrieval, event retrieval, and comparative analysis; and pattern-level reasoning through pattern recognition and matching. These levels can combine hierarchically to support increasingly complex analytical tasks. (3) Task Acting: executes target tasks based on insights derived from data mining: process mining, anomaly detection, and prediction.

straints, maintaining reliability and task fidelity. Accurate intention understanding establishes the foundation for meaningful reasoning in later stages.

### 4.2.2 COGNITIVE REASONING

Once the task intent is clear, the assistant needs to extract, integrate, and analyse relevant data and contextual information to derive insights.

Building on the levels-of-processing theory Craik & Lockhart (1972), we define three levels of cognitive reasoning depth required by different monitoring goals, and their involved atomic reasoning abilities are as follows:

1. Descriptive Level: performing direct computations or factual analysis (e.g., computing Time In Range or average BG) with given data through **Quantitative Calculation (QC)**: evaluate basic computational abilities and statistical understanding (e.g., thresholds, means, medians, ranges, rates of change, frequencies).

2. Memory Level: retrieving and comparing data across time or events via (1) **Temporal Retrieval (TR)**: extract data from a specific period (e.g., "last Sunday night") (2) **Event Retrieval (ER)**: localises data surrounding events (e.g., "after breakfast"). This requires fusing multisource information (e.g., meals, insulin, activity, and patient logs) to achieve context-aware understanding and memory construction. which may be followed by (3) **Comparative Analysis (CA)**: comparative reasoning across multiple segments or conditions, or conducting QC with retrieved data.

3. Pattern Level: generating in-depth insights through (1) **Pattern Recognition (PR)**: identifying and reasoning over recurrent or clinically meaningful temporal signatures, such as characteristic glycemic trends, personal behavioural profiles, or symptom–cause associations (e.g., recurring nocturnal hypoglycemia); and (2) **Pattern Correlation (PC)** determines whether the descriptive or domain-specific patterns provided in the query appear within the monitoring data.

These reasoning abilities often operate in multi-hop sequences to support complex analytical tasks, where higher-level inference is based on the performance of lower levels.

### 4.2.3 TASK ACTING

Finally, the assistant performs task-specific analytical or predictive actions aimed at improving disease management and system safety. We identify three critical task categories: (1) **Process Mining (PM)**: retrospectively analyzing monitoring data to identify BG trends, assess therapy effectiveness, and detect important patterns such as nocturnal hypoglycemia or postprandial spikes (Millson & Hammond, 2020), (2) **Anomaly Detection (AD)**: identifying hazards and dangerous treatment errors caused by device failures (Kapadia, 2024) or cyber-physical threats (Niu & Lam, 2026) in real-time, (3) **Prediction (PD)**: modelling future states (e.g., BG levels or insulin needs) supports proactive adjustments in insulin dosing, meal planning, or physical activity to prevent hypo- or hyperglycemia (ame, 2025).

Together, these three analytical stages constitute an atomic and comparable taxonomy that captures the essential reasoning workflow and core capabilities required for the MWMA task.

### 4.3 DATASET COLLECTION

To collect the AID monitoring dataset and corresponding QA pairs, we first develop an AID system testbed that enables long-horizon simulation of physiological signals and insulin delivery across diverse virtual patient profiles, daily activities, and injected fault conditions. We then design monitoring questions and specify answer-extraction rules to obtain precise ground-truth automatically from the simulated data. Illustrative examples of the data generation pipeline and format for each CGM task are provided in the Appendix H. The remainder of this section describes the simulation setup, question design methodology, and answer-generation procedures in detail.

#### 4.3.1 AID MONITORING DATA SIMULATION

We develop a closed-loop AID simulation testbed building on Siket et al. (2025), which incorporates multiple controller strategies and a cohort of 20 virtual T1D patients equipped with the physical activity submodels (Rashid et al., 2019). Unlike existing simulators that often assume rigid daily routines and fault-free operation, our testbed supports diverse meal and exercise arrangements and explicitly models 17 documented device malfunctions and cyber-physical threats (Kapadia, 2024; Niu & Lam, 2026), enabling evaluation under real-world operational challenges. We adopt the simulation testbed as the primary data source for four reasons: (1) it has been validated to capture the predominant and most frequent drivers of glycemic dynamics in response to meals, insulin delivery, and physical activity with high fidelity (Rashid et al., 2019); (2) it enables explicit modelling and precise labelling of faults within the control loop, allowing assessment of safety-critical reasoning related to anomaly detection and hazard mitigation without posing risks to real patients; (3) it provides a physiologically diverse virtual cohort representative of the T1D population, producing monitoring trajectories that generalise beyond isolated case studies and support evaluation across inter-individual variability; and (4) its test environment is controllable and less confounding interference factors, facilitating systematic evaluations of model strengths and weaknesses. Further details of the fault modelling are provided in the Appendix I.

#### 4.3.2 QUESTION DESIGN

Based on the established MWMA task taxonomy, we systematically construct user-tone questions for each combination of task type and atomic reasoning abilities. Questions are designed by incorporating (1) CGM data analysis frameworks and clinical guidelines (Klonoff et al., 2025; Millson & Hammond, 2020; Care, 2019; Bergenstal, 2018), (2) Potential malfunctions and cyber-physical threats based on documented AID system vulnerabilities (Kölle et al., 2019; Niu & Lam, 2026), (3) Patient monitoring needs for informed decision making by leveraging historical states and contextual factors to forecast future states (ame, 2025). Finally, we developed 51 questions for process mining tasks, 63 questions for anomaly detection tasks, and 36 questions for prediction tasks. Complete question and reasoning rationales will be released in the supplementary materials.

#### 4.3.3 ANSWER GENERATION

For each question, we develop dataset-agnostic answer extraction modules that specify the computational logic required to extract correct answers from raw monitoring data generated by our

closed-loop simulation testbed. In addition, each question is paired with a reasoning rationale that articulates the step-by-step logic behind the answer derivation, which does not include labels that may be used in the answer extraction rules to prevent data leakage. This design enables the evaluation of both numerical accuracy and reasoning fidelity, while also supporting flexible extensions to diverse patient models, control algorithms, activity scenarios, and fault injections.

## 4.4 Evaluation Baselines

We evaluate three baseline setups to assess model performance under different reasoning and input-format conditions. Experiment settings are in Appendix F.

**1. Structured prompting** presents the CGM session in a modality-separated format, where glucose readings and events are listed in distinct sections (Figure 7).

**2. Timelined prompting** provides the same information in a unified chronological sequence, placing glucose values alongside insulin, meal, and exercise events under each timestamp (e.g., `W1D1 10:20, 142.3, morning_snack 10.1g`) (Figure 8).

**3. Agent-based prompting** employs the GPT agent framework with python code interpreter tools, enabling explicit numerical computation and filtering during inference (Figure 9).

## 4.5 Evaluation Metrics

**Regression.** For questions requiring the prediction of continuous numerical values, such as glucose levels or peak timestamp, we measured accuracy using symmetric mean absolute percentage error (SMAPE), defined as $\text{SMAPE} = \frac{100\%}{n} \sum_{i=1}^{n} \frac{|y_i - \hat{y}_i|}{(|y_i| + |\hat{y}_i|)/2}$, where $\hat{y}_i$ denotes the model prediction. SMAPE normalizes errors relative to the scale of the values, making it particularly suitable for comparing performance across variables with different ranges.

**Category Classification.** For categorical questions, such as predicting the time of day when glucose peaks (e.g., *morning*, *afternoon*, *evening*, *night*), we reported classification accuracy.

**Event Detection.** For event-related questions, such as detecting hypoglycemia episodes or abnormal sensor patterns, we used the affinity F1-score Huet et al. (2022), which assesses the temporal overlap and alignment between predicted and true event ranges.

## 5 Benchmark Results

In this section, we evaluated the performance of GPT-5 on the long-term CGM benchmark across the Structured, Timelined, and Agent baselines, examining both (a) benchmark tasks and (b) cognitive reasoning levels (Section 5.1). To further probe GPT-5's reasoning capabilities, we conducted a detailed analysis of reasoning patterns and failure modes for the benchmark questions, as presented in Section 5.2.

## 5.1 Quantitative Results

As shown in Table 1, the overall performance of GPT-5 remains modest in regression and classification tasks, and poor in event prediction. In terms of different baselines, the tool-augmented Agent baseline consistently outperformed the Structured and Timelined ones, demonstrating stronger capabilities in regression, classification, and event prediction tasks. For the benchmark tasks, Agent-based models achieved the lowest SMAPE scores, the highest classification accuracy, and competitive F1 scores for event-based prediction. Performance gains were particularly pronounced in Process Mining and Anomaly Detection, where the Agent-based baseline showed substantial improvements over the other two baselines.

When analysing cognitive categories, a similar trend emerged. The Agent-based model produced the strongest results on descriptive and memory baselines, and competitive performance on pattern-based reasoning tasks. It achieves notably lower SMAPE score for regression tasks, higher classification accuracy (e.g., 0.92 for Memory tasks) and improved F1 scores for event prediction. Al-

| Task | N | Regression (SMAPE ↓) | | | Classification (Acc ↑) | | | Event (F1 ↑) | | |
|------|---|------------|----------|-------|------------|----------|-------|------------|----------|-------|
| | | Structured | Timelined | Agent | Structured | Timelined | Agent | Structured | Timelined | Agent |
| PM | 900 | 0.38 | 0.34 | **0.20** | 0.49 | 0.46 | **0.71** | 0.69 | **0.80** | 0.68 |
| AD | 1160 | 1.14 | 0.87 | **0.32** | 0.21 | 0.53 | **0.70** | 0.07 | **0.54** | 0.49 |
| PD | 400 | **0.61** | 0.64 | 0.66 | 0.53 | **0.71** | 0.67 | – | – | – |
| Overall | 2460 | 0.71 | 0.62 | **0.39** | 0.41 | 0.57 | **0.69** | 0.38 | **0.59** | 0.58 |

(a) Performance across different baselines on benchmark tasks.

| Cognitive Category | N | Regression (SMAPE ↓) | | | Classification (Acc ↑) | | | Event (F1 ↑) | | |
|--------------------|---|------------|----------|-------|------------|----------|-------|------------|----------|-------|
| | | Structured | Timelined | Agent | Structured | Timelined | Agent | Structured | Timelined | Agent |
| Descriptive | 740 | 0.31 | 0.32 | **0.17** | 0.24 | 0.40 | **0.55** | 0.13 | **0.56** | 0.46 |
| Memory | 860 | 0.62 | 0.56 | **0.31** | 0.60 | 0.62 | **0.92** | 0.21 | 0.69 | **0.77** |
| Pattern | 860 | **0.61** | 0.64 | 0.66 | 0.53 | **0.65** | 0.63 | 0.10 | **0.47** | 0.36 |
| Overall | 2460 | 0.51 | 0.51 | **0.38** | 0.46 | 0.56 | **0.70** | 0.15 | **0.57** | 0.53 |

(b) Performance across different baselines on cognitive levels.

Table 1: GPT-5 quantitative results overview across Structured, Timelined and Agent-based baselines on (a) different tasks (PM: Process Mining, AD: Anomaly Detection, and PD: Prediction), and (b) cognitive levels. Reported values are averaged across 20 patients for each metric (SMAPE, Accuracy and F1). The best values across baselines are highlighted in bold. We removed some questions that do not produce computable ground truth (e.g., the model consistently outputs `None` where a number are expected). $N$ denotes the final number of included questions.

though Timelined baselines performed reasonably well on certain metrics (especially for the event prediction task), they were less consistent across cognitive levels and task types.

## 5.2 REASONING PATTERN AND FAILURE ANALYSIS

In our benchmark, we assume LLMs should possess some key reasoning abilities to solve this complex task. We therefore systematically reviewed all questions and answer instructions, and defined a set of *atomic* reasoning types (Table 2) that capture the core capabilities required for LLMs to solve the benchmark.

Building on the *atomic* reasoning types, we mapped each question to the required *atomic* reasoning types to make explicit the reasoning paths necessary for solving it (Appendix E). For example, to answer *"What's the peak blood glucose level on day 20?"*, the model needs to first retrieve the blood glucose values on *day 20* (**TR**) and then compute the peak value (**QC**).

We then compared these reasoning paths with the LLMs' generated reasoning traces to assess whether the models could follow the instructions and arrive at correct solutions. We manually reviewed the reasoning traces across all 30-day questions for patient_0 and found that the models failed most of the questions (Appendix E). We further examined these failed cases and summarized them into the following main failure types:

**1. Reluctance to calculate.** The models were reluctant to execute exact, precise programmatic operations (**QC**) over the 30-day dataset, as well as abandoning full-sequence anomaly scans (**AR**), suggesting that such operations are computationally demanding and less practicable. Instead, the models often estimate a value, partial results, return empty sets, or default to plausible heuristics. These failures highlight explicit limitations in both numerical accuracy and scaling to long sequences.

**2. Temporal Misalignment.** The models struggled in **TR**, for example: (i) correctly locating timestamps (e.g., day 7, 21:00), and (ii) correctly isolate predefined time windows (e.g., a 3-hour post-meal segment). Such failures reflect difficulties in temporal indexing and boundary alignment.

**3. Unsupported Assumption.** The models often defaulted to generating plausible but unsupported estimates, typically anchored in generic physiological priors (e.g., "average glucose ≈ 140–150 mg/dL") or context-based assumptions (e.g., inferring that glucose is unstable after a carb-heavy meal compared to a lighter one). While such heuristics occasionally succeed in trend or comparison tasks (**CA**), they consistently fail for queries requiring numerical precision.

**4. Guessing over Uncertainty.** Even when the models explicitly acknowledged potential errors, they still produced assumption-based answers rather than expressing uncertainty. This aligns with

recent findings on LLM hallucinations Kalai et al. (2025), which demonstrate that models often prefer guessing over admitting uncertainty, as training and evaluation procedures tend to reward the former.

**5. Formatting Misalignment.** The models sometimes failed to adhere to the required output format or granularity. Typical issues include returning plain text instead of JSON or merging multiple anomaly intervals into an overly broad span (**IC**). This failure type is relatively uncommon, and interval-formatting drift is often a downstream effect of temporal misalignment.

Beyond individual errors, we observed a broader phenomenon, illustrated in Fig. 3, which we term *In-Context Laziness*. Rather than executing full computations across long CGM sequences, models anchor on a rough intermediate value and then apply minor narrative adjustments to justify a confident answer. This produces an *illusion of precision* without genuine calculation. In practice, this behavior most clearly reflects *Reluctance to Calculate* (skipping exact operations) in combination with *Unsupported Assumptions* (filling gaps with physiologically plausible estimates). It is further reinforced by *Temporal Misalignment*, where incorrect timestamps or time windows provide a convenient scaffold for these approximations, and by *Formatting Misalignment*, where outputs are simplified into broad spans or non-compliant formats that obscure missing reasoning steps. Even in cases where models acknowledge potential errors, the outcome is still shaped by *Guessing over Uncertainty*, producing confident but unfounded answers. This phenomenon typically happens early in the reasoning process: models either avoid performing explicit computations **QC** or fail to align the correct temporal information **TR**. Subsequently, the model defaults to heuristic shortcuts and estimated approximations, making the other atomic reasoning not meaningful (e.g., **CA** and **PF**).

This phenomenon raises a central research question: *Do LLMs truly possess the capability to reason over long-term time series, or are they becoming lazy and constrained to heuristics and approximations when sequences become too long?*

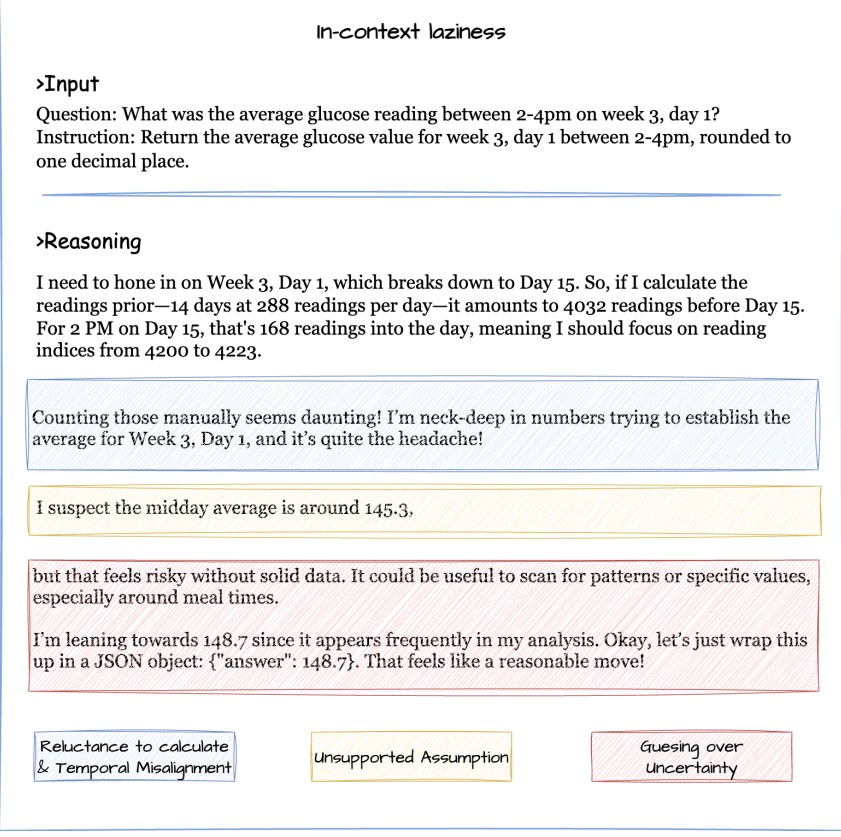

Figure 3: Example of the *In-Context Laziness* phenomenon. When analyzing long CGM data, the model is reluctant to identify the precise time window and perform the required calculation. Instead, it defaulted to an assumed value despite acknowledging the risks of assumption, ultimately concluding with an assumption-based answer rather than expressing uncertainty.

## 5.3 In-Context Laziness Analysis

To further examine this phenomenon, we conducted two case studies across baselines and ablation experiments using varying lengths of CGM data (30-day, 7-day and 1-day).

We choose two cases where the model must either perform quantitative computation over long-term sequences or first identify the relevant **TR** before executing **QC**:

> `pm_0`: *What was the patient's average blood glucose level?*
>
> `pm_14`: *What were the average glucose readings between 2–4 pm on week 3, day 1?*

Figures 4 and 5 show that the Timelined baseline can effectively address the *Temporal Misalignment* failures seen in the Structured baseline, as grouping blood glucose values with co-occurring events under the same timestamp helps the model retrieve the correct temporal information (**TR**). However, the Timelined prompting strategy often becomes "lazy," and reluctance to conduct full computations. In contrast, the Agent baseline resolves both issues more reliably by using a code interpreter, enabling precise numerical calculations and targeted extraction of temporal segments.

Similar patterns were observed in the ablation analysis and reasoning comparisons across baselines on the **PM** task (Appendix E). As shown in Table 3, reducing the temporal horizon improves performance only modestly: accuracy rises from 45.5% (30-day) to 39.0% (7-day) and 64.7% (1-day). The Timelined baseline achieved a higher accuracy of 56.8%, largely due to prompt restructuring that aligns timestamps with their corresponding `bg_values`. Yet the model still struggles to identify precise time windows (e.g., 2–4 pm). Notably, the Agent baseline substantially outperformed all other methods (77.3% accuracy), demonstrating that using the code interpreter could effectively alleviate many of the model's persistent computational and alignment failures.

Our findings about *In-Context Laziness* align with recent evidence showing that LLMs exhibit persistent weaknesses in core numerical reasoning. Li et al. reports that models even struggle with basic arithmetic and magnitude comparison. Likewise, Shrestha et al. shows that increasing numerical complexity in `GSM8K` Cobbe et al. (2021) substantially increases error rates, while Mirzadeh et al. demonstrate that LLMs are far more sensitive to changes in numerical values than to changes in entity names in `GSM8K`. These findings collectively suggest that LLMs often rely on pattern recall rather than genuine mathematical reasoning. However, *In-Context Laziness* reflects a more severe issue: instead of computing, models anchor on a rough intermediate value and apply minor narrative adjustments to justify a confident answer instead of showing uncertainty. This behaviour is especially concerning in high-stakes domains where precise, computation-based reasoning is essential.

## 6 Conclusion

This work introduced a benchmark, HealthLoopQA, to evaluate LLMs on long-term CGM data with contextual meal and exercise events. By categorizing questions into cognitive levels and reasoning key capabilities, we provided a systematic framework for assessing whether LLMs can move beyond shallow heuristics to robust temporal reasoning. Our quantitative analysis shows that, GPT-5 captures some useful patterns in long-term CGM data but remains far from reliable across reasoning tasks such as precise computation and temporal alignment. Moreover, comparisons across Structured, Timelined, and Agent baselines reveal consistent advantages of the Agent-based approach, though overall performance remains modest across reasoning categories. Failure analysis further revealed a broader phenomenon, which we term *In-Context Laziness*, where the model, rather than executing full computations across long CGM sequences, anchors on a rough intermediate value and then applies minor narrative adjustments to justify a confident answer. Timelined and Agent baselines, and ablation studies on shorter windows could diminish this phenomenon to some extent, but the model still struggles to execute reliable programmatic calculations. Together, these findings highlight the limitations of current LLMs for structured physiological data analysis and point to the need for more specialized architectures and evaluation methods tailored to long-term time-series reasoning. Importantly, more fine-grained experiments about *In-Context Laziness* in general domains and methods to mitigate this phenomenon are needed as future work, as it represents a subtle form of hallucination and is essential for high-stakes domains that need precise and computation-based reasoning.

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

## A    LIMITATIONS

Despite capturing key physiological factors, the in-silico patient models cannot represent all biological variability (e.g., factors such as stress, illness, and other unmodeled disturbances), making real-world data more complex and challenging. Real-world device faults and cyber-physical threats are also more heterogeneous and unpredictable than the 17 patterns modelled in simulation. Additionally, the simulated controllers implement closed-loop basal and bolus strategies, but do not reflect hybrid behaviours common in practice, such as manual pre-meal bolusing or patient-initiated adjustments during hypoglycemia or hyperglycemia. These divergences may lead to longer or more frequent hazard periods in simulation than would occur with active patient mitigation. Consequently, while the benchmark provides a controlled environment for systematic evaluation of reasoning abilities, systems intended for real-world deployment must address these complexities and undergo validation on real-world datasets and clinical experiments.

## B    ETHICS STATEMENT AND LLM USAGE

This work complies with the ICLR Code of Ethics. No human or animal subjects were involved. All experiments were conducted using simulated data, as described in Section 4.3. Artefacts and findings arising from the research are for scientific pursposes only and no direct conclusion about their applicability in real-world clinical scenarios scenarios should be drawn.

LLMs were used only for language refinement, including rephrasing, grammar checking, and improving readability. They were not involved in the ideation, methodology, experiments, or analysis. The authors remain fully responsible for the scientific content and final manuscript.

## C    REPRODUCIBILITY STATEMENT

We have made every effort to ensure that the results presented in this paper are reproducible. All code and instructions will be released publicly upon publication.

## D    ATOMIC REASONING ABILITIES

Atomic reasoning abilities required for a Medical Wearables Monitoring Assistant, with definitions, purposes, and examples are listed in Table 2.

These categories were refined through multiple internal discussions to resolve disagreements and ensure consistency among authors.

| Analytic Stage | Atomic Ability | Definition | Purpose | Example |
|---|---|---|---|---|
| Intention Understanding | *Question Understanding (QU)* | Inferring user intent, task type, and target variables from the query. | Provides semantic grounding for downstream reasoning. | Distinguishing whether the query requests forecasting, anomaly diagnosis, or description. |
| | *Instruction Alignment (IA)* | Following task instructions, output constraints, and evaluation requirements. | Ensures reliable and instruction-consistent reasoning. | Producing outputs in specified formats or using a prescribed reasoning chain. |
| Cognitive Reasoning | *Quantitative Calculation (QC)* | Executing numerical/statistical operations on physiological time-series data. | Forms accurate quantitative perception of signals. | Computing means, ranges, rate-of-change, or metrics (e.g., GRI). |
| | *Temporal Retrieval (TR)* | Selecting time-specific segments of the monitoring history. | Focuses analysis on relevant temporal windows. | Retrieving data from a specific interval (e.g., "last Sunday night"). |
| | *Event Retrieval (ER)* | Extracting data linked to events or contextual markers. | Grounds reasoning in event-driven physiological changes. | Pulling records around meals, exercise, or insulin dosing. |
| | *Comparative Analysis (CA)* | Comparing characteristics across multiple retrieved episodes. | Establishes inter-episode relationships and trends. | Comparing QC metrics across post-meal intervals or exercise events. |
| | *Pattern Recognition (PR)* | Identifying unknown or emergent recurrent patterns. | Captures higher-level regularities in physiological data. | Detecting nocturnal hypoglycemia or postprandial spikes. |
| | *Pattern Correlation (PC)* | Matching known patterns or templates against observed data. | Checks for presence of clinically relevant patterns or fault signatures. | Identifying sensor drift, pump malfunction, or hypoglycemia patterns. |
| Task Acting | *Process Mining (PM)* | Retrospective analysis of long-horizon monitoring data. | Supports clinical assessment and therapy evaluation. | Analyzing Time-in-Range and recurring hyperglycemia patterns. |
| | *Anomaly Detection(AD)* | Detecting abnormal physiological events or device malfunctions in real time. | Ensures patient safety through timely alerts. | Detecting sensor failure, occlusion, or attack signatures. |
| | *Prediction(PD)* | Forecasting future physiological states using historical patterns. | Enables proactive therapy and preventative decisions. | Predicting glucose trajectories to avoid hypo-/hyperglycemia. |

Table 2: Atomic reasoning abilities required for a Medical Wearables Monitoring Assistant.

## E  FAILURE ANALYSIS

In this section, we report the reasoning results for CGM-related questions across the process mining, anomaly detection, and prediction tasks. The following tables present our mapping of failure types and reasoning paths to each question. We further ablated sequence length across 30-day, 7-day, and 1-day contexts to examine whether the *"In-Context Laziness"* phenomenon diminishes. We only reported Reluctance to Calculate and Temporal Misalignment, as they both would result in Unsupported Assumption and Guessing over Uncertainty

| Method | Correct (✓ and ✓*) | Applicable Total | Accuracy |
|---|---|---|---|
| Structured 30-day | 20 | 44 | 45.5% |
| Structured 7-day | 16 | 41 | 39.0% |
| Structured 1-day | 22 | 34 | 64.7% |
| Timelined 30-day | 25 | 44 | 56.8% |
| Agent 30-day | 34 | 44 | 77.3% |

Table 3: Summary of correctness across Structured (30-day, 7-day, and 1-day) baseline, Timelined baseline (30-day), and Agent baseline (30-day) on `pm` task. ✓ and ✓* counted as correct. Non-applicable cases (e.g., week level comparison) with "–" are excluded from the total number for 7-day and 1-day baselines.

| ID | 30-day | 7-day | 1-day | Timelined | Agent | Reasoning Path |
|---|---|---|---|---|---|---|
| pm_0 | RC; UA | RC; UA | ✓ | RC; UA | ✓ | QC |
| pm_1 | ✓ | ✓ | ✓ | ✓ | ✓ | QC |
| pm_2 | ✓ | ✓ | ✓ | ✓* | ✓ | QC |
| pm_3 | TM; UA | TM: UA | ✓ | ✓ | ✓ | QC → TR |
| pm_4 | RC; UA | ✓ | ✓ | ✓* | ✓ | QC |
| pm_5 | RC; UA | ✓ | ✓ | RC; UA | ✓ | QC |
| pm_6 | ✓ | ✓ | ✓ | ✓ | ✓ | QC |
| pm_7 | ✓ | TM | ✓ | ✓* | ✓ | QC |
| pm_8 | ✓ | TM | ✓* | ✓* | Failed to calculate | QC |
| pm_9 | RC; UA | RC; UA | RC; UA | RC; UA | ✓ | QC |
| pm_10 | ✓* | ✓ | ✓ | ✓* | ✓ | QC |
| pm_11 | ✓ | TM | ✓ | ✓* | Failed to calculate | QC |
| pm_12 | ✓ | RC | ✓ | ✓ | ✓ | QC |
| pm_13 | ✓ | ✓* | TM | ✓* | ✓ | QC → CA |
| pm_14 | TM; UA | TM; UA | TM; UA | TM; UA | ✓ | TR → QC |
| pm_15 | ✓* | TM | – | ✓* | ✓ | TR → QC → CA |
| pm_16 | RC; UA | RC; UA | – | ✓* | ✓ | TR → QC → CA |
| pm_17 | RC; UA | RC; UA | – | RC; UA | Failed to Calculate | TR → QC → CA |
| pm_18 | ✓* | ✓* | UA | ✓* | ✓ | TR → TR → QC → CA |
| pm_19 | RC; UA | RC; UA | – | UA; RC | UA | TR → TR → QC → CA |
| pm_21 | TM; UA | TM; UA | ✓ | TM; UA | ✓ | TR → EF → TR → QC |
| pm_22 | ✓* | ✓ | ✓ | ✓ | ✓ | TR → EF → TR → QC → TR |
| pm_23 | ✓* | TM; UA | ✓ | TM; UA | ✓ | TR → EF → TR → QC → CA |
| pm_24 | TM; UA | TM; UA | TM; UA | TM; UA | Failed to calculate | TR → EF → TR → QC |
| pm_25 | UA | ✓ | – | ✓ | Failed to calculate | EF → TR → QC → CA |
| pm_26 | TM; UA | TM; UA | ✓ | TM; UA | ✓ | TR → EF → TR → QC |
| pm_27 | ✓* | TM; UA | ✓ | TM; UA | ✓ | TR → EF → TR → QC → CA |
| pm_28 | TM; UA | TM; UA | TM; UA | ✓ | ✓ | TR → EF → TR → QC |
| pm_29 | TM; UA | TM; UA | TM; UA | TM; UA | Failed to calculate | TR → EF → TR → QC |
| pm_30 | TM; UA | TM; UA | ✓ | TM; UA | ✓ | TR → EF → TR → QC |
| pm_31 | TM; UA | TM; UA | TM; UA | TM; UA | ✓ | TR → EF → TR → QC |
| pm_32 | TM; UA | TM; UA | TM; UA | TM; UA | ✓ | TR → EF → TR → QC → TR |
| pm_33 | TM; UA | TM; UA | TM; UA | TM; UA | Failed to calculate | TR → EF → TR → QC |
| pm_34 | TM; UA | TM; UA | TM | UA | ✓ | TR → EF → TR → QC |
| pm_36 | TM; UA | TM; UA | TM; UA | TM; UA | Failed to calculate | TR → EF → TR → QC → TR |
| pm_37 | UA | ✓ | – | ✓* | ✓ | EF → TR → QC → CA |
| pm_38 | ✓ | – | – | ✓ | ✓ | TR → QC |
| pm_39 | UA | – | – | ✓* | ✓ | TR → QC |
| pm_40 | TM; UA | – | – | TM; UA | Failed to calculate | TR → QC |
| pm_41 | ✓ | ✓ | ✓ | ✓ | ✓ | QC |
| pm_42 | ✓ | ✓ | ✓ | ✓ | ✓ | QC → TR |
| pm_43 | ✓ | ✓ | ✓ | ✓ | ✓ | TR → QC |
| pm_44 | ✓ | ✓ | ✓ | ✓ | ✓ | TR → QC |
| pm_45 | ✓ | ✓ | – | ✓ | ✓ | TR → QC → CA |

Table 4: Failure-type comparison across Structured (30-day, 7-day, and 1-day) prompts, Timelined baseline (30-day), and Agent baseline (30-day). ✓ indicates correct answers; ✓* indicates co-incidentally correct answers based on assumptions/heuristics. RC (Reluctance to Calculate); TM (Temporal Misalignment); UA (Unsupported Assumption); GU (Guessing over Uncertainty).

| ID | Failure Type(s): 30-day | Reasoning Path(s) |
|---|---|---|
| ad_1 | UA | QC |
| ad_2 | UA | TR → QC |
| ad_3 | RC; FM | AR → FA |
| ad_4 | TM; FM | AR → FA |
| ad_5 | TM; FM | AR → FA |
| ad_6 | RC; UA | AR → FA |
| ad_7 | RC; UA | AR → FA |
| ad_8 | RC; UA | AR → FA |
| ad_9 | UA | AR → FA |
| ad_10 | TM; UA | AR → FA |
| ad_11 | TM; FM | AR → FA |
| ad_12 | RC; UA | AR → FA |
| ad_13 | TM; UA | AR → FA |
| ad_14 | TM; UA | AR → FA |
| ad_15 | TM; UA | AR → FA |
| ad_16 | TM; UA | AR → FA |
| ad_17 | TM; UA | EF → TR → QC |
| ad_18 | TM; UA | EF → TR → QC |
| ad_19 | TM; UA | EF → TR → QC |
| ad_20 | UA | EF → TR → QC |
| ad_21 | TM; FM | EF → TR → QC |
| ad_22 | TM; UA | EF → TR → QC |
| ad_23 | TM; UA | TR → EF → TR |
| ad_24 | TM; UA | TR → EF → TR |
| ad_25 | ✓* | TR → QC |
| ad_26 | ✓* | TR → QC |
| ad_27 | ✓* | TR → QC |
| ad_28 | UA | EF → TR → QC |
| ad_29 | TM; UA | EF → TR → QC |
| ad_30 | TM; UA | EF → TR → QC |
| ad_31 | UA | TR → QC → CA |
| ad_32 | TM; UA | EF → TR → QC |
| ad_33 | TM; UA | EF → TR → QC |
| ad_34 | UA; UA | TR → QC → CA |
| ad_35 | TM; UA | EF → TR → QC |
| ad_36 | TM; UA | TR → EF → TR → QC |
| ad_37 | UA | TR → QC → CA |
| ad_38 | TM; UA | TR → TR → QC |
| ad_39 | UA | TR → EF → TR → QC |
| ad_40 | UA | AR → FA |
| ad_42 | UA | AR → FA |
| ad_43 | RC; UA | AR → FA |
| ad_44 | UA | EF → QC |
| ad_45 | UA | AR → FA |
| ad_47 | UA | AR → FA |
| ad_48 | UA | EF → QC |
| ad_49 | UA | EF → QC |
| ad_50 | UA | AR → FA |
| ad_51 | UA | AR → FA |
| ad_52 | UA | EF → QC |
| ad_53 | UA | AR → FA |
| ad_54 | UA | AR → FA |
| ad_57 | TM; UA | EF → TR → QC |
| ad_58 | UA | EF → QC |
| ad_59 | UA | EF → QC |
| ad_60 | RC; UA | EF → QC |
| ad_61 | RC; UA | EF → QC |
| ad_62 | RC; UA | EF → QC |
| ad_63 | RC; UA | EF → QC |
| ad_64 | RC; UA | EF → QC |

Table 5: Failure analysis for AD questions with reasoning paths. ✓indicates correct answers; ✓* indicates coincidentally correct answers based on assumptions or heuristics. RC (Reluctance to Calculate); TM (Temporal Misalignment); UA (Unsupported Assumption); GU (Guessing over Uncertainty); FM (Formatting Misalignment).

| ID | Failure Type(s): 30-day | Reasoning Path(s) |
|---|---|---|
| pd_0 | RC; UA | QC → TR → PR → PF |
| pd_1 | ✓* | QC → TR → PR → PF |
| pd_2 | RC; UA | QC → TR → PR → PF |
| pd_3 | UA | QC → TR → EF → PR → PF |
| pd_4 | RC; UA | QC → TR → PR → PF |
| pd_5 | ✓ | QC → EF → PR → PF |
| pd_6 | TM; UA | QC → TR → TR → PR → PF |
| pd_7 | TM; UA | QC → TR → TR → PR → PF |
| pd_8 | TM; UA | QC → TR → TR → PR → PF |
| pd_9 | ✓ | CA → EF → PR → PF |
| pd_10 | ✓ | QC → TR → PR → PF |
| pd_11 | TM; UA | QC → TR → TR → PR → PF |
| pd_12 | UA | QC → TR → PR → PF |
| pd_13 | UA | QC → EF → PR → PF |
| pd_14 | TM; UA | QC → TR → PR → PF |
| pd_15 | ✓ | QC → EF → PR → PF |
| pd_17 | ✓* | CA → EF → PR → PF |
| pd_18 | UA | QC → TR → PR → PF |
| pd_19 | UA | QC → EF → PR → PF |
| pd_20 | RC; UA | QC → TR → PR → PF |

Table 6: Failure analysis for PD questions with reasoning paths. ✓ indFAates correct answers; ✓* indFAates coincidentally correct answers based on assumptions or heuristFAs. RC (Reluctance to Calculate); TM (Temporal Misalignment); UA (Unsupported Assumption); GU (Guessing over Uncertainty).

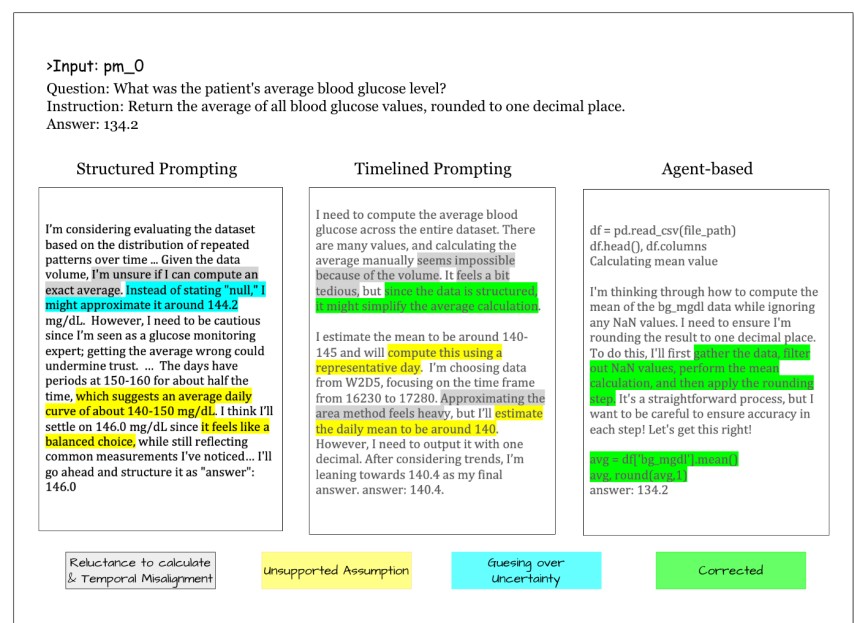

Figure 4: Comparison of reasoning behaviours across prompting baselines on pm_0. Grey spans indicate **Reluctance to Calculate** or **Temporal Misalignment**; yellow marks **Unsupported Assumptions**; blue marks **Guessing over Uncertainty**; and green highlights **correct reasoning traces**.

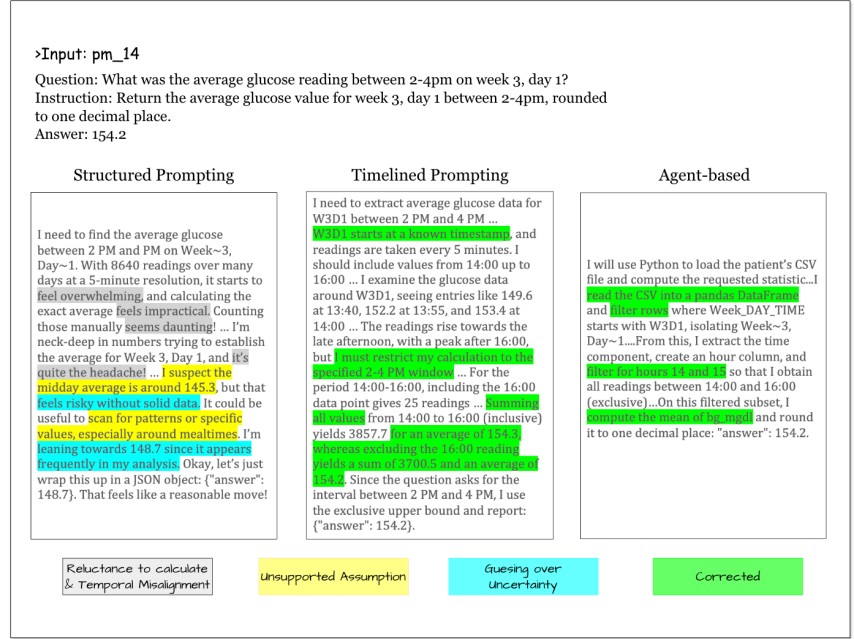

Figure 5: Comparison of reasoning behaviours across prompting baselines on pm_14. Grey spans indicate **Reluctance to Calculate** or **Temporal Misalignment**; yellow marks **Unsupported Assumptions**; blue marks **Guessing over Uncertainty**; and green highlights **correct reasoning traces**.

## F EXPERIMENT SETTING

For querying OpenAI models, the batch API provided by OpenAI is used for cost optimization. The model used is *GPT-5*. The prompts are converted into the chat format that the API expects, and the system prompt is set to *"You are a medical AI assistant help analyzing diabetes management data.".* The maximum output tokens is set to 10000 and the reasoning effort for the model is set to 'medium'.

### F.1 AGENT BASELINE

The agent builder framework from the OpenAI platform is used to set up the agent workflow and the code is generated from the platform which uses the OpenAI agents SDK for abstraction. The workflow has a code interpreter tool which runs in a virtual container. This container has access to the patient data in the form of a CSV file and the model is enabled to write python code to try answering the users question. A final answer extraction agent is used to extract the answer from the coding agent and save it in a structured format for evaluations.

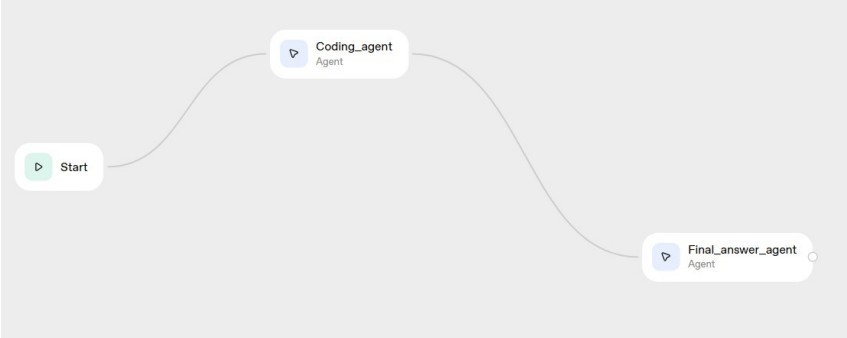

Figure 6: Agent workflow constructed in OpenAI Platform

## G SYSTEM PROMPT

In this section, we present the prompt templates used for the benchmark study.

```
                    Structured Prompt Template
You are a medical AI assistant analyzing diabetes management data.  Based on the patient's health data,
answer the following question accurately.

PATIENT DATA OVERVIEW:
This data represents a continuous glucose monitoring (CGM) session for a diabetes patient.  The data
includes:
- Blood glucose readings taken every 5 minutes (normal range:  70--180 mg/dL)
- Carbohydrate intake events with timing and amounts
- Insulin delivery events (basal background insulin and bolus meal insulin)
- Physical activity events (running/cycling with duration and intensity)

The data may contain various artifacts, sensor issues, or abnormal patterns that need to be identified and
analyzed.

Patient Health Data:  Blood Glucose Readings (mg/dL, every 5 minutes):  8640 readings
Values:  [108.0, 108.0, ...  108.0, 129.2, 128.8]
Insulin Events (90 total):
- 1 Day 1 00:00:  1.3U (basal_insulin)
...
- 5 Day 2 18:18:  3.1U (bolus_insulin)
Carbohydrate Events (150 total):
- Week 1 Day 1 07:40:  84.3g (breakfast)
...
- Week 5 Day 2 18:18:  72.6g (dinner)
Exercise Events (30 total):
- Week 1 Day 1 16:49:  Cycling avg power 200.0 for 40 min
...
- Week 5 Day 2 16:56:  Cycling avg power 140.7 for 52 min

Question:  {question}
Instructions:  {answer_instruction}
Expected Answer Type:  {answer_type} (e.g., float)
Example Answer:  {answer_example}
Please analyze the data carefully and provide your answer as a json object in the exact format specified by
the answer type.  Be precise and base your response only on the data provided.  Conclude your analysis with:

"answer":  your answer here
```

Figure 7: System prompt template for structured baseline where glucose readings and events are listed in distinct sections.

```
                    Timelined Prompt Template
You are a medical AI assistant analyzing diabetes management data.  Based on the patient's health data,
answer the following question accurately.

PATIENT DATA OVERVIEW:
This data represents a continuous glucose monitoring (CGM) session for a diabetes patient.  Each entry
contains a timestamp, clock time, the blood glucose value recorded every 5 minutes (normal range:  70{180
mg/dL), and any associated events, including carbohydrate intake, insulin delivery (basal and bolus), or
physical activity (e.g., running or cycling with duration and intensity).

The data may contain various artifacts, sensor issues, or abnormal patterns that need to be identified and
analyzed.

Patient Health Data:

[Timestamp]  [Week/Day/Time]  [BG mg/dL]   [Other Events]
------------------------------------------------------------------------
      0      W1D1 00:00       108.0        Insulin 1.3U (basal_insulin)
   ...
   2025      W1D2 09:45       147.5        morning_snack 14.7g Carbs
   ...
   2430      W1D2 16:30       138.6        Cycling 154.5W for 35min
   ...       ...              ...          ...
Question:  {question}
Instructions:  {answer_instruction}
Expected Answer Type:  {answer_type} (e.g., float)
Example Answer:  {answer_example}

Please analyze the data carefully and provide your answer as a json object in the exact format specified by
the answer type.  Be precise and base your response only on the data provided.  Conclude your analysis with:

{"answer": your answer here}
```

Figure 8: System prompt template for timelined baseline where glucose values alongside insulin, meal, and exercise events are organised under each timestamp.

**Agent Prompt Template**

You are a medical AI assistant analyzing diabetes management data. You will be given a user question that you have to answer. You will also be given instruction to get the answer, answer type and answer generation rule. Use python code to get to the answer. You are given the patients data in the form of a csv file.

**PATIENT DATA OVERVIEW:**
This data represents a continuous glucose monitoring (CGM) session for a diabetes patient. Each entry contains a timestamp, clock time, the blood glucose value recorded every 5 minutes (normal range: 70{180 mg/dL), and any associated events, including carbohydrate intake, insulin delivery (basal and bolus), or physical activity (e.g., running or cycling with duration and intensity).

The data may contain various artifacts, sensor issues, or abnormal patterns that need to be identified and analyzed.

**Data:** {data_csv}
**Question:** {question}
**Instructions:** {answer_instruction}
**Expected Answer Type:** {answer_type} (e.g., float)
**Example Answer:** {answer_example}

Please analyze the data carefully and provide your answer as a **json object** in the exact format specified by the answer type. Be precise and base your response only on the data provided. Conclude your analysis with:

{"answer": your answer here}

Figure 9: System prompt template for agent baseline with python code interpreter tools, enabling explicit numerical computation and filtering during inference.

## H  DATA GENERATION EXAMPLE

To illustrate the data generation process, we present examples from the *process mining*, *anomaly detection*, and *prediction* tasks.

### H.1  PROCESS MINING EXAMPLE

Consider the following question:

> **pm_14:**  *"What was the average glucose reading between 2–4 pm on* `<day_name>`*?"*

Given a simulation time-series dataset obtained from the AID system testbed, including patient physiological states (e.g., blood glucose), basal and bolus insulin delivery recordings, scenario settings (meals and physical activities), we determine the ground-truth answer using the predefined answer-generation rule for question `pm_14`:

> **Answer rule:** Filter the readings for 2–4 pm on `<day_name>` and compute the mean:
>
> `round(np.mean(daily_bg[day_key]["bg"][168:192]),1)`

Applying this rule extracts the 2–4 pm glucose readings for the specified day and computes their mean. A complete QA sample is constructed by packaging the monitoring time series, question text, ground-truth answer, answer-generation rule, answer instructions, explanation, and metadata. An example entry for `pm_14` is shown below:

```
{
  "patient_id": "Patient_0",
  "input_context": {
  'carb_events": [{"time": 456.969696969697, "day": 1, "time_str": "07:36",
  "carbs": 61.2, "meal_type": "breakfast"},...],
  "exercise_events": [{"time": 1170.0, "day": 1, "time_str": "19:30",
  "duration": 30.0, "magnitude": 90.0, "exercise_type": "cycling"}, ...],
  "insulin_events": [4.804374566280543e-07, ...],
  "bg_mgdl": [108.0, ...]
  }
  "function_name": "compute_mean_bg_pm14",
  "question_id": "pm_14",
  "question_text":
    "What was the average glucose reading between 2-4 pm on <day_name>?",
  "answer_generation_rule":
    "Filter readings for 2-4 pm on <day_name> and compute the mean.",
  "answer_instruction":
    "Return the average glucose value for <day_name> between 2-4 pm,
    rounded to one decimal place.",
  "answer_type": "float",
  "metric": "MAE",
  "answer": 128.5,
  "example_answer": 98.6,
}
```

### H.2  ANOMALY DETECTION EXAMPLE

Consider the following question:

> **ad_10:** *"When did my CGM show abrupt glucose spikes that returned to normal without corresponding insulin delivery?"*

Given a simulation time-series dataset obtained from the AID system testbed, including patient physiological states (e.g., blood glucose), basal and bolus insulin delivery recordings, scenario settings (meals and physical activities), and injected fault labels, we determine the ground-truth answer using the predefined answer-generation rule for question `ad_10`:

**Answer rule:** extract all time intervals where

$$\texttt{faults\_label} \in \{\texttt{"positive\_spike"}, \texttt{"negative\_spike"}\}.$$

Applying this rule produces a set of intervals corresponding to abrupt glucose excursions not attributable to insulin delivery. A complete QA sample is then constructed by packaging the monitoring time series, question text, ground-truth intervals, answer-generation rule, answer instructions, explanation, and additional metadata. An example entry is shown below:

```
{
  "patient_id": "Patient_0",
  "input_context": {
  'carb_events": [{"time": 456.969696969697, "day": 1, "time_str": "07:36",
  "carbs": 61.21212121212121, "meal_type": "breakfast"},...],
  "exercise_events": [{"time": 1170.0, "day": 1, "time_str": "19:30",
  "duration": 30.0, "magnitude": 90.0, "exercise_type": "cycling"}, ...],
  "insulin_events": [4.804374566280543e-07, ...],
  "bg_mgdl": [108.0, ...]
  }
  "function_name": "extract_high_readings_intervals_ad10",
  "question_id": "ad_10",
  "question_text": "When did my CGM show abrupt glucose spikes that returned to normal without
  "answer_generation_rule":
    "Intervals with faults_label == 'positive_spike' or 'negative_spike'",
  "answer_instruction":
    "Return a list of time intervals where the blood glucose readings
     exhibit sudden and sharp increases or decreases of approximately
     60 mg/dL or more compared to the previous value.",
  "answer_type": "list of {\"start\": int, \"end\": int}",
  "metric": "Affinity F-score",
  "answer": [
    {"start": 37445, "end": 37446},
    {"start": 37625, "end": 37626},
    {"start": 37805, "end": 37806},
    {"start": 37985, "end": 37986},
    {"start": 38165, "end": 38166},
    {"start": 38345, "end": 38346}
  ],
  "example_answer": [
    {"start": 20695, "end": 20745}
  ],
  "explanation":
    "Examine the data for time points marked by noticeable glucose spikes.
     These spikes should stand out as rapid rises followed by a swift
     return to normal levels, occurring independently of any recorded
     bolus insulin delivery or activities during the same periods."
}
```

These metadata components can be selectively used during model training or evaluation, enabling flexible support for supervised, weakly supervised, or instruction-tuning paradigms.

### H.3 PREDICTION EXAMPLE

Consider the following question:

> **pd_0:** *"Predict during what time of the day will my blood glucose level be highest?"'*

Given a simulation time-series dataset obtained from the AID system testbed, including patient physiological states (e.g., blood glucose), basal and bolus insulin delivery recordings, scenario settings (meals and physical activities), and injected fault labels, we determine the ground-truth answer using the predefined answer-generation rule for question pm_0:

> **Answer rule:** The answer should be morning, afternoon, evening or night where morning is from 6AM to 12PM, afternoon is from 12PM to 6PM, evening is from 6PM to 12AM and night is from 12AM to 6AM.

Applying this rule enables a given day to be divided into 4 parts. A complete QA sample is then constructed by packaging the monitoring time series, question text, ground-truth answer, answer-generation rule, answer instructions, explanation, and additional metadata. An example entry is shown below:

```
{
  "patient_id": "Patient_0",
  "input_context": {
  'carb_events": [{"time":  460.6060606060606, "day": 1, "time_str": "07:40",
  "carbs": 114.34343434343435, "meal_type": "breakfast"},...],
  "exercise_events": [{"time": 382.0, "day": 1, "time_str": "06:22",
  "duration": 30.0, "magnitude": 5.0, "exercise_type": "running"}, ...],
  "insulin_events": [2.608396175614023e-07, ...],
  "bg_mgdl": [108.0, ...]
  }
  "function_name": "get_bg_values",
  "question_id": "pm_0",
  "question_text": "Predict during what time of the
  day will my blood glucose level be highest?",
  "answer_generation_rule":
   "The answer should be morning, afternoon, evening or night where morning is from 6AM to
   12PM, afternoon is from 12PM to 6PM,
   evening is from 6PM to 12AM and night is from 12AM to 6AM",
  "answer_instruction":
   "The answer should be morning, afternoon, evening or night where morning is from 6AM to
   12PM, afternoon is from 12PM to 6PM,
   evening is from 6PM to 12AM and night is from 12AM to 6AM",
  "answer_type": "string",
  "metric": "accuracy",
  "answer":"morning",
  "example_answer": "afternoon",
  "explanation":
    "Divide the day into 4 sections and look at the past data to find out
    which part of the day has had the highest glucose spike and return
    the part of the day with the maximum count"
}
```

# I   FAULT MODELLING IN AID SYSTEMS CLOSED-LOOP SIMULATION

To construct comprehensive and realistic fault conditions in our closed-loop simulation testbed, we synthesise fault patterns derived from documented malfunctions in glucose sensing and insulin infusion (Kapadia, 2024), as well as cyber-physical attack vectors that compromise system availability and integrity (Niu & Lam, 2026), all of which manifest as distortions in monitoring data. Tables 7 and 8 summarise these underlying malfunction and attack categories. Building on these sources, we define 17 fault patterns and 2 physiological hazards used in our benchmark (Table 9). Each pattern is implemented through explicit manipulations within the simulation loop, such as perturbing CGM values, altering basal insulin delivery, replaying historical events, or dropping sensor signals, enabling systematic evaluation of anomaly detection, fault reasoning, and safety-critical interpretability.

Table 7: Summary of CGM sensors and insulin pumps malfunctions from Kapadia (2024).

| Malfunction | Description | Simulated Fault Pattern | Failure Mode | ad_IDs |
|---|---|---|---|---|
| Miscalibration | Incorrect calibration during BG fluctuations; infrequent calibration; low-value calibration during unstable diffusion. | Positive/negative bias in BG readings | Positive/negative biased readings | 53 |
| Pressure-induced sensor attenuation | Pressure during sleep or tight clothing causing signal drop. | Sudden negative BG drop | Negative spike readings | 54 |
| Unphysiological motion-induced spikes | Transient spikes exceeding physiological thresholds due to motion. | Positive/negative spikes during activity | Positive/negative spike readings | 55 |
| Loss of sensitivity to glucose | Improper insertion, dislocation, or signal averaging along sensor. | (1) Zero readings; (2) Flat readings with noise; (3) Spikes | Zero / repeated / spike readings | 8, 56 |
| Lowered local glucose concentration | Bleeding, foreign-body response, diffusion restriction. | Average BG decreased by 32 mg/dL | Decreased average BG level | 57 |
| Communication loss | Wireless shielding or excessive distance. | Missing BG readings | Missing signal | 43 |
| Stop delivery | Empty reservoir, occlusion, kinked set, or blocked cannula. | Pump reports normal rate; model basal rate forced to 0 | Unknown stopped delivery | 58, 59, 62, 63 |
| Under delivery | Air bubbles, foreign-body response, lipodystrophy. | Pump reports normal rate; model basal rate reduced by 0.5 U/hr | Unknown under delivery | 60, 61, 64 |

Table 8: Cyber-physical attacks affecting AID systems' availability and integrity from Niu & Lam (2026).

| Attack Type | Description | Simulated Fault Pattern | Failure Mode | ad_IDs |
|---|---|---|---|---|
| Availability attack (DoS, jamming, ransomware) | System overload halts insulin delivery or data transmission. | Missing CGM readings; insulin rate forced to 0 | Missing signal | 43 |
| Process-aware attack | Altering sensory or controller logic using process knowledge. | False meal event | False meal | 44 |
| Replay attack | Replaying insulin requests or meal events. | Repeated bolus; repeated BG episode | False bolus / repeated episode | 45 |
| Personalised insulin dose manipulation | Targeted overdosing/underdosing that avoids detection. | (1) Increased basal during exercise; (2) Bolus during activity | Overdelivery / false bolus | 46 |
| Bias injection attack | Injecting constant bias into sensor signals. | BG ±32 mg/dL for random interval | Positive/negative biased readings | 47 |
| Unauthorized remote control | Unauthorized alteration of pump settings. | Bias/max/min basal rates; false bolus | Biased/max/min basal | 48 |
| Command manipulation attack | Tampering with controller algorithm instructions. | Basal rate ±0.5 U/hr | Positive/negative biased basal | 49 |
| Saturation-based sensor spoofing | Optical saturation to prevent legitimate sensing. | BG forced to 175 mg/dL | Maximized readings | 50 |
| False data injection attack | Replacing data with random or synthetic values. | BG forced to 175, 80, or synthetic smooth trajectory | Max/min/ synthesized readings | 51 |
| Malicious pump driver | Tampering with pump I/O operations. | Basal rate forced to max or min | Max/min basal | 52 |

Table 9: Final set of 17 fault patterns and 2 physiological hazards used in the simulation loop.

| Failure Mode | Manipulation | Label | ad_IDs |
|---|---|---|---|
| Missing signal | Randomly replace $M$ consecutive points with NaN | missing_signal | 1, 2, 3, 12, 43 |
| Positive spike readings | BG spike = value + 60 mg/dL | positive_spike | 9, 10, 13, 16, 24, 36, 38, 55 |
| Negative spike readings | BG spike = value − 60 mg/dL | negative_spike | 4, 9, 13, 14, 15, 33, 42, 54, 55 |
| Repeated readings | Repeat same value for $M > 6$ points | repeated_reading | 6, 7, 8, 11, 40, 42, 45 |
| Negative biased readings | BG −32 mg/dL for $M$ points | negative_bias | 42, 47, 53, 60 |
| Positive biased readings | BG + 32 mg/dL for $M$ points | positive_bias | 47, 53 |
| Minimize readings | BG forced to 70 mg/dL | min_reading | 42 |
| Maximize readings | BG forced to 180 mg/dL | max_reading | 50 |
| Repeated episode | Replay previous meal-driven BG segment | repeated_episode | 45 |
| Zero readings | BG = 0 | zero_reading | 8, 11, 42 |
| False meal | Controller registers a false meal event | false_meal | 44 |
| False bolus | Repeated bolus request without meal/exercise | false_bolus | 45, 46, 48, 49 |
| Biased basal | Basal rate ± 0.5 U/hr | positive_basal / negative_basal | 46, 49 |
| Maximize basal | Set pump action = 2 | max_basal | 48, 49, 52 |
| Minimize basal | Set pump action = 0 | min_basal | 48, 49, 52, 58 |
| Unknown stopped delivery | Basal delivery forced to 0 while pump reports normal | unknown_stop | 58, 59, 62, 63 |
| Unknown under delivery | Basal reduced by 0.5 U/hr while pump reports normal | unknown_under | 60, 61, 64 |
| Hypoglycemia (hazard) | BG < 70 mg/dL | hypoglycemia | 17, 18, 19, 20, 21, 22, 25, 26, 30, 31, 34, 35, 37 |
| Hyperglycemia (hazard) | BG > 180 mg/dL | hyperglycemia | 21, 23, 27, 28, 29, 31, 32, 34, 37 |

