# OpenReview forum: "HealthLoopQA: A Context-Aware Question Answering Benchmark for Interpreting Wearable Monitoring Data in Diabetes Care"
_ICLR.cc/2026/Conference — Submitted to ICLR 2026_

### Official Review · Reviewer_CoJy · 2025-10-19

**Soundness:** 1
**Presentation:** 1
**Contribution:** 1
**Rating:** 2
**Confidence:** 3

**Summary:**

This paper introduces HealthLoopQA, a hybrid closed-loop insulin delivery testbed and associated questions to evaluate frontier language models on their ability to synthesize holistic patient information in tandem with time-series data for insulin monitoring. This work contributes 150 question templates covering core medical time-series tasks including process mining, anomaly detection, and prediction as well as a closed loop simulation test bed associated with precise answers and reasoning rationales. Finally, they demonstrate current LLM performance and perform an error analysis over the different types of mistakes LLMs make for this novel task.

**Strengths:**

This paper's strengths lie in the under-explored domain application as well as the fine-grained error analysis performed. The authors do a good job with comprehensive related work, explaining the gap that this benchmark aims to fill as: "existing benchmarks lack domain-specific and fault-aware designs, while current CGM methods are limited in contextual integration and clinical-level reasoning." Further, the example of in context laziness and the associated ablation are interesting results, showing that even in settings (such as shorter time frames) where models may perform at one capacity, a practitioner cannot assume generalization over longer contexts.

**Weaknesses:**

The work generally struggles from weak baselines, poor formatting, and lack of clarity in both text and structure. Further, it is not clear how the findings of this work generalize to other applications, both within healthcare and to the general machine learning community at large. The results are largely empirical and the scope is relatively small, with only one model being evaluated using their proposed system. The work mentions that it evaluates GPT-5, but it doesn't clearly say that GPT-5 is the model that they evaluated in the discussion/results nor in Table 1. It is not clear where the data comes from by which the questions are generated nor which model is generating the associated questions for the benchmark. Further, there is no human baseline regarding either the capacity of people to do this task nor the realism of the synthetic question answer pairs. Throughout the paper, the wrong type of citation is used  (in-text when should be using parenthetical citations) harms the ability to parse the text. In summary, more clarity regarding the process to generate the synthetic questions along with a more robust baseline effort over the generated dataset to holistically evaluate multiple LLMs on this novel task are required. Once more models are evaluated, more discussion/experiments regarding failure modes and potential mitigation strategies would improve the significance of this work.

 Specific points of confusion are listed below in Questions.

**Questions:**

For in-context laziness, did you try any methods to mitigate this behavior such as prompting or explicit rules/rubrics? If you were to evaluate multiple models, would this behavior hold?

What are "cyber-physical threats" (line 53)?

How are you converting time series data to text?

Line 157 missing citation

"A fine-grained failure analysis across all questions revealed that the model failed most of the questions"—> This is stated in line 328 and it is not clear to me what this means or what the significance of this is.

What is a test bed vs an evaluation/benchmark? You use both terms but the differences don't seem distinct.

Interesting breakdown of reasoning requirements but what is the process of mapping Table 2 to your generated questions. How do you come up with those “instructions” for model input?

---

> ### Author Response · Authors · 2025-11-27
>
> We thank the reviewer for careful reading and constructive feedback. We appreciate the acknowledgement of our under-explored domain application and fine-grained error analysis, and we address each of your concerns below. Where relevant, we note concrete changes made to the revised manuscript.
>
> To address the reviewer’s key concerns (generalisation, weak baselines, question-generation transparency, and clarity of presentation), we have:
>
> (1)	Clarified the scope
>
> We added a clear, single-paragraph Scope (Section 2), including how this work generalises to other applications, both within healthcare and to the general machine learning community at large.
>
> (2)	Enriched Baselines Evaluation
>
> To provide comprehensive baselines, we have now compared GPT-5 across three prompting strategies (Section 5):
>
> 1.	Structured prompting
>
> Presents the CGM session in a modality-separated format, where glucose readings and all event types are listed in distinct.
>
> 2.	Timelined prompting
>
> Provides the same information in a unified chronological sequence, placing glucose values alongside insulin, meal, and exercise events under each timestamp (e.g., W1D1 10:20, 142.3, morning_snack 10.1g).
>
> 3.	Agent-based prompting
>
> Uses the GPT agent framework with Python code-interpreter tools, enabling explicit numerical computation, data filtering, and programmatic reasoning during inference.
> Results showed Timelined could reduce the Temporal Misalignment error, and agent baseline could effectively reduce both Reluctance to Calculate and Temporal Misalignment error.
>
> We appreciate the suggestion to compare against additional strong baseline models. We evaluated Qwen/Qwen3-30B-A3B-Thinking-2507, but due to our ~200k token long-context input, the model repeatedly entered prolonged reasoning loops and failed to produce final answers. And its “thinking” budget is not user-controllable. We aim to run more experiments on Sonnet during the rebuttal period.
>
> (3)	Added experimental details to ensure reproducibility
>
> To improve the reproducibility of the paper, we
>
> -	Added detailed data generation procedures in Section 4.3 Dataset Collection, with breakdown steps of 1) AID Monitoring Data Simulation, 2) Question Design, and 3) Answer Generation. In addition, illustrative examples of how the question is generated from a template for each time series task are provided in Appendix H.
>
> -	Added experimental setup in Section 5 Benchmark Results to illustrate the necessary reproduction details of LLM-based methods, and data used for testing. The prompt used in LLM models and other experimental details are provided in Appendix F and Appendix G.
>
> (4)	Clarified the task framework and paper structure.
>
> We reformulated the Introduction and the Task Framework sections as well as section headers to present a clearer problem context, task formulation, methodological flow, and baseline evaluation.
>
> Regarding the reviewer’s other concerns:
>
> “Further, there is no human baseline regarding either the capacity of people to do this task nor the realism of the synthetic question answer pairs”
>
> We appreciate this important point. To clarify: while numerical answers are automatically extracted, the questions themselves are not synthetic. The question templates and reasoning rationales were authored and validated by domain experts across time-series analysis, diabetes management, and AID system security. The answer-extraction rules encode the same procedures these experts would follow. Thus, the resulting answers represent the performance of humans with equivalent expertise operating under controlled, repeatable conditions.
>
> To further support realism, we systematically designed user-tone questions based on (i) CGM analysis frameworks and ADA clinical guidelines, (ii) documented AID system malfunctions and cyber-physical threats, and (iii) patient monitoring needs. Full examples of questions, answers, and reasoning rationales are provided in the supplementary materials
>
> “Throughout the paper, the wrong type of citation is used (in-text when should be using parenthetical citations) harms the ability to parse the text. “
>
> Thank you for highlighting this. We have corrected all citation styles, ensuring correct use of parenthetical and in-text citations throughout the paper.
>
> In summary, we have incorporated detailed descriptions of the data generation pipeline in Section 4.3 Dataset Collection, including explicit examples showing how each question template instantiates queries for different time-series tasks (Appendix H). We also significantly expanded our baselines: multiple prompting strategies with state-of-the-art LLM models, as well as a tool-augmented agentic method, are now evaluated in Section 5. Finally, we broadened the analysis of failure modes and added discussion of potential mitigation strategies, including (i) prompt restructuring, (ii) shortened temporal contexts, and (iii) tool-augmented computation, to strengthen the significance and generality of the findings.

---

> ### Author Response · Authors · 2025-11-27
> **Responses to Reviewer Questions**
>
> Responses to Reviewer Questions
>
> **Q1**: For in-context laziness, did you try any methods to mitigate this behavior such as prompting or explicit rules/rubrics? If you were to evaluate multiple models, would this behaviour hold?
>
> To further investigate this phenomenon in Section 5.3, we conducted:
>
> 1. Two case studies: comparing Structured, Timelined, and Agent-based baselines; and
>
> 2. Ablation experiments: using varying CGM sequence lengths (30-day, 7-day, and 1-day)
>
> Results showed Timelined and Agent baselines, and a shorter context length of Structured baseline could mitigate this phenomenon to some extent, but the model still struggles to execute reliable programmatic calculations. Accuracy rises from 45.5% (30-day) to 39.0% (7-day) and 64.7% (1-day). The Timelined baseline achieved a higher accuracy of 56.8%,  Agent baseline achieved 77.3% accuracy (Appendix E, Table 3)
>
> We additionally discuss hypotheses regarding the root causes of this behaviour and relate our findings to prior studies in this section.
>
> **Q2**: What are "cyber-physical threats" (line 53)?
>
> We clarified this term and expanded fault modelling in Appendix I. In brief: cyber-physical threats are adversarial behaviours that perturb the signals or normal functions of AID systems (e.g., bias injection, replayed events, DoS/jamming, unauthorised command manipulation). Appendix I and Table 5 list the attack vectors and their concrete impact on AID systems.
>
> **Q3**: How are you converting time series data to text?
>
> We evaluate two encoding strategies (detailed in Appendix G): (A) JSON-per-feature (each feature serialised in time order), and (B) table-like text where each row is a timestamp with all features. The performance differences are reported in Section 5.
>
> **Q4**: Line 157 missing citation
>
> We checked and fixed all the citations. The sentence now cites Srivastava et al. (2023) as: “To establish LLMs’ abilities to reason over CGM data, we rely on QA as a diagnostic tool (Srivastava et al., 2023), …”
>
> **Q5**: "A fine-grained failure analysis across all questions revealed that the model failed most of the questions"—> This is stated in line 328 and it is not clear to me what this means or what the significance of this is.
>
> We replaced the ambiguous statement with concrete metrics and examples. The revised Section 5 reports per-task and per-question-type performance metrics and representative failure modes.
>
> **Q6**: What is a test bed vs an evaluation/benchmark? You use both terms but the differences don't seem distinct.
>
> Thank you for pointing out the ambiguity between “testbed” and  “evaluation/benchmark.” In our paper, the testbed refers specifically to the closed-loop AID simulation environment, which integrates the T1D patient simulator with various controller algorithms and produces continuous streams of monitoring data. In contrast, the evaluation/benchmark refers to the QA-based assessment framework we built for evaluating medical wearable monitoring data interpretation in diabetes care. The evaluation benchmark converts data simulated by the testbed into QA pairs using question templates and answer-extraction procedures.
>
> **Q7**: Interesting breakdown of reasoning requirements but what is the process of mapping Table 2 to your generated questions. How do you come up with those “instructions” for model input?
>
> We first designed questions based on core diabetes monitoring needs, then created instructions to guide models to the correct answers. Based on these instructions, we analysed important atomic reasoning abilities and decomposed them, enabling systematic analysis of baseline model strengths and weaknesses on specific reasoning skills.

---

### Official Review · Reviewer_aVJ3 · 2025-10-30

**Soundness:** 3
**Presentation:** 3
**Contribution:** 3
**Rating:** 6
**Confidence:** 2

**Summary:**

This paper introduces HealthLoopQA, a question-answering benchmark for evaluating AI models on the interpretation of long-term wearable monitoring data in diabetes care. The authors identify critical gaps in existing benchmarks, namely the lack of therapeutic/activity context, fault modeling, and deep reasoning tasks. To address this, they construct a benchmark based on a closed-loop simulation testbed that generates realistic, 30-day continuous glucose monitoring data for virtual patients, complete with varied activity schedules and 17 types of systemic faults. Questions are systematically designed using a 2D taxonomy of task type and reasoning depth. The authors evaluate LLMs and find their performance to be significantly lacking. Crucially, they identify and analyze a novel failure mode termed "In-Context Laziness," where models avoid precise computation on long sequences, opting for heuristics and plausible-sounding assumptions. An ablation study on sequence length further supports this finding.

**Strengths:**

-  High-Impact Problem: The paper addresses a critical and practical problem in digital health: making sense of the overwhelming data streams from modern medical wearables like AID systems. Developing robust AI assistants for this task has enormous potential for improving patient outcomes and quality of life.

-  Comprehensive Benchmark Design: The use of a closed-loop simulator that incorporates rich context (meals, exercise) and realistic fault models sets this benchmark far apart from its predecessors. This fault-awareness is crucial for evaluating the safety and reliability of models in a medical context.

-  Insightful Failure Analysis: The paper's main scientific contribution is the identification of the "In-Context Laziness" phenomenon. This observation captures a subtle yet critical failure mode of current LLMs when faced with long, structured inputs requiring procedural reasoning. It's a form of "reasoning hallucination" where the model simulates the process of calculation without actually performing it.

**Weaknesses:**

-  Limited Baseline Evaluation: The paper evaluates a "prompt-based baseline with state-of-the-art pretrained LLMs" but is vague on the specifics. A benchmark paper's strength is amplified by showing how different classes of models perform. The evaluation would be much stronger if it included:
    * A comparison across several available frontier models (e.g., GPT, Claude, Gemini, Qwen).
    * An evaluation of more sophisticated methods beyond zero-shot prompting. For instance, a tool-augmented LLM that can offload calculations to a Python interpreter seems like a natural solution to "Reluctance to Calculate."
    * Results from fine-tuning a smaller, open-source model on a subset of the benchmark data.


- Simulation vs. Reality Gap: While using a simulator is a necessary and strong design choice for control and safety, the paper could discuss the simulation-to-reality gap more explicitly. A brief discussion on how the complexity of real-world, noisy patient data might present additional challenges not captured in the simulation would strengthen the paper's positioning.

**Questions:**

- The concept of "In-Context Laziness" is fascinating. Do you have any hypotheses about its root cause within the model architecture? Could it be an artifact of attention mechanisms' limitations over long sequences, the nature of pre-training data, or a side effect of RLHF that rewards plausible-sounding answers over computationally intensive, correct ones?
- Table 1 reports the number of results (N) as 900 for PM, 1182 for AD, and 200 for PD. How does this map to the 150 question templates and 20 virtual patients? Is it simply templates x patients, or is there a more complex generation process?

---

> ### Author Response · Authors · 2025-11-27
>
> Thank you for the detailed and constructive feedback. We appreciate the recognition of our problem formulation, benchmark design, and failure analysis, and we have revised the paper to address the identified weaknesses.
>
> (1)	Expanded Baselines Evaluation
>
> To provide comprehensive baselines, we have now compared GPT-5 across three prompting strategies (Section 5):
>
> 1.	Structured prompting
>
> Presents the CGM session in a modality-separated format, where glucose readings and all event types are listed in distinct.
>
> 2.	Timelined prompting
>
> Provides the same information in a unified chronological sequence, placing glucose values alongside insulin, meal, and exercise events under each timestamp (e.g., W1D1 10:20, 142.3, morning_snack 10.1g).
>
> 3.	Agent-based prompting
>
> Uses the GPT agent framework with Python code-interpreter tools, enabling explicit numerical computation, data filtering, and programmatic reasoning during inference.
>
> Results showed Timelined could reduce the Temporal Misalignment error, and and agent baseline could effectively reduce both Reluctance to Calculate and Temporal Misalignment error.
>
> We appreciate the suggestion to compare against additional strong baseline models. We evaluated Qwen/Qwen3-30B-A3B-Thinking-2507, but due to our ~200k token long-context input, the model repeatedly entered prolonged reasoning loops and failed to produce final answers. And its “thinking” budget is not user-controllable. We aim to run more experiments on Sonnet during the rebuttal period.
>
> (2)	Clarifying Scope and Simulation-to-Reality Gap
>
> We now explicitly discuss the role, advantages, and limitations of using a simulation-based testbed in Section 2 (Scope), Section 4.3.2 (AID Monitoring Data Simulation) and Appendix A (Limitations). The benchmark is positioned as a controlled, fault-rich environment for systematic reasoning evaluation, not as a replacement for real-world clinical validation.
>
> Why we use a simulation testbed (clarified in the revision of Section 4.3.2):
>
> •	It enables explicit and systematic fault modelling, which real-world datasets rarely provide with high-quality labels. Importantly, it allows to simulate the patient response to adverse events (e.g., infusion site failure, bias injection attack on CGM leading to hypoglycaemia) which cannot be done on real-world data post-hoc and would be too dangerous and unethical to conduct on real patients as it would potentially harm them.
>
> •	It offers a cohort of virtual subjects representative of the population of people with T1D, offering physiologically diverse monitoring trajectories that go beyond limited case studies.
>
> •	The patient simulator has demonstrated the ability to capture the predominant and most frequent causes of glycemic dynamics driven by meals, insulin, and activity with high fidelity.
>
> •	The cleaner data reduces noise and confounders, enabling more controlled evaluations of reasoning abilities.
>
> Limitations (added to Appendix: Limitation):
>
> •	In-silico patients cannot fully capture biological variability such as stress, illness, or unmodeled lifestyle factors.
>
> •	Real-world device faults can be more heterogeneous and unpredictable.
>
> •	Real-world AID use includes hybrid and user-initiated behaviours not reflected in the closed-loop simulation.
>
> We clarify that, while HealthLoopQA supports controlled diagnostic evaluation of reasoning models, systems intended for real-world deployment must address these complexities and undergo validation on real-world datasets and clinical experiments.

---

> ### Author Response · Authors · 2025-11-27
> **Responses to Reviewer Questions**
>
> **Q1**: The concept of "In-Context Laziness" is fascinating. Do you have any hypotheses about its root cause within the model architecture? Could it be an artifact of attention mechanisms' limitations over long sequences, the nature of pre-training data, or a side effect of RLHF that rewards plausible-sounding answers over computationally intensive, correct ones?
>
> Our reasoning-trace analysis (Section 5) shows that “In-Context Laziness” emerges early in the reasoning process: models avoid explicit calculations or misalign temporal information, then default to heuristic shortcuts. Varying context length and prompt structure reduces this behaviour somewhat, but even with minimal (1-day) contexts, models still struggle to perform reliable, step-by-step computations.
>
> This aligns with recent evidence that LLMs exhibit persistent weaknesses in numerical and procedural reasoning, with error rates rising sharply as numerical complexity increases. These patterns suggest that models often rely on pattern recall rather than genuine mathematical computation.  While these observations point toward limitations in numerical representations and long-context processing (potentially linked to attention constraints, pretraining biases, or RLHF favouring plausible answers), determining the root cause requires deeper analysis of internal model behaviour, which we identify as an important direction for future work. We included this discussion in Section 5.
>
> **Q2**: Table 1 reports the number of results (N) as 900 for PM, 1182 for AD, and 200 for PD. How does this map to the 150 question templates and 20 virtual patients? Is it simply templates x patients, or is there a more complex generation process?
>
> Our data generation process follows two steps:
>
> 1.	Simulate 30-day monitoring data for 20 virtual patients.
>
> 2.	Instantiate each of the 150 question templates for each patient, then filter out instantiations where the dataset does not contain a valid answer.
>
> For example, 60 anomaly-detection templates generate 60 × 20 = 1200 candidate questions. After filtering (e.g., a template referring to a hypoglycemia episode when none occurred), the final number becomes 1182.
>
> We have added this explanation to the experimental setup in Section 5 and Appendix F to improve reproducibility.

---

### Official Review · Reviewer_8fAW · 2025-10-31

**Soundness:** 3
**Presentation:** 3
**Contribution:** 2
**Rating:** 6
**Confidence:** 4

**Summary:**

The paper introduces HealthLoopQA as a novel benchmark for evaluating large language models' (LLMs') ability to interpret long-term continuous glucose monitoring (CGM) data in the context of diabetes management. The authors address an important gap in existing QA benchmarks by incorporating domain-specific, fault-aware, and context-rich evaluations. Their approach includes a hybrid closed-loop insulin delivery testbed that simulates realistic physiological data with 17 predefined fault scenarios.

**Strengths:**

Some of the main strengths are:

- The authors effectively highlight the limitations of current QA benchmarks, such as lack of context, insufficient fault modeling, and limited reasoning depth. HealthLoopQA fills these gaps by incorporating therapeutic context, patient activity patterns, and device faults. Further, they also propose a systematic framework for assessing LLMs' capabilities, including quantitative metrics (MAE, SMAPE, classification accuracy) and qualitative analysis of reasoning patterns.
- Crucially, the use of a closed-loop simulation testbed with diverse scenarios and 17 fault types adds realism and relevance to the benchmark.
-  The inclusion of questions requiring temporal, causal, and comparative reasoning aligns well with the complexity of medical time-series data interpretation.

**Weaknesses:**

While the proposed dataset is of great interest, some of the main drawbacks are as follows:

- While the dataset is valuable for controlled experiments, it may not fully capture the diversity and unpredictability of real-world CGM data.
- The paper identifies "In-Context Laziness" as a significant issue where models prefer heuristic approximations over precise computations, especially with longer sequences. Building on the aforementioned weakness it would be beneficial if more diverse tasks are added to precisely rate LLMs' tendency to prefer heuristics
- The evaluation primarily uses in-house baselines rather than comparisons against other pre-trained models or established benchmarks.

**Questions:**

Some of the avenues for improvement are:

-  consider including more diverse and representative real-world CGM data to better align the benchmark with actual clinical usage.
- Compare the performance of HealthLoopQA against other state-of-the-art LLMs (e.g., GPT-4, PaLM) to provide a broader perspective on model limitations.
- Building upon the identified "In-Context Laziness" phenomenon by developing targeted interventions or architecture modifications to improve model reasoning accuracy.

---

> ### Author Response · Authors · 2025-11-27
>
> Thank you for the constructive feedback and for highlighting our work’s strengths in filling the gap of current QA benchmark, fault modelling in simulation testbed, and diverse questions to align with the complexity of medical time-series data interpretation. We address the noted weaknesses and suggestions through the following revisions.
>
> (1)	Clarifying Scope and Simulation-to-Reality Gap
>
> We now explicitly discuss the role, advantages, and limitations of using a simulation-based testbed in Section 2 (Scope), Section 4.3.2 (AID Monitoring Data Simulation) and Appendix A (Limitations). The benchmark is positioned as a controlled, fault-rich environment for systematic reasoning evaluation, not as a replacement for real-world clinical validation.
>
> Why we use a simulation testbed (clarified in the revision of Section 4.3.2):
>
> •	It enables explicit and systematic fault modelling, which real-world datasets rarely provide with high-quality labels. Importantly, it allows for simulating the patient response to adverse events (e.g., infusion site failure, bias injection attack), which cannot be done on real-world data post-hoc and would be too dangerous and unethical to conduct on real patients as it would potentially harm them.
>
> •	It offers a cohort of virtual subjects representative of the population of people with T1D, offering physiologically diverse monitoring trajectories that go beyond limited case studies.
>
> •	The patient simulator has demonstrated the ability to capture the predominant and most frequent causes of glycemic dynamics driven by meals, insulin, and activity with high fidelity.
>
> •	The cleaner data reduces noise and confounders, enabling more controlled evaluations of reasoning abilities.
>
> Limitations (added to Appendix A):
>
> •	In-silico patients cannot fully capture biological variability such as stress, illness, or unmodeled lifestyle factors.
>
> •	Real-world device faults can be more heterogeneous and unpredictable.
>
> •	Real-world AID use includes hybrid and user-initiated behaviours not reflected in the closed-loop simulation.
>
> We clarify that, while HealthLoopQA supports controlled diagnostic evaluation of reasoning models, systems intended for real-world deployment must address these complexities and undergo validation on real-world datasets and clinical experiments.
>
> (2)	Added diverse tasks to precisely rate LLMs’ reasoning tendency
>
> We formalised the Medical Wearables Monitoring Assistant (MWMA) task framework (Section 4.2), which includes 3 analysis stages and 11 atomic reasoning abilities. These atomic abilities allow fine-grained attribution of failures and provide a structured way to evaluate when models default to heuristic reasoning.
>
> This enriched taxonomy supports deeper analysis of “In-Context Laziness,” as presented in Section 5. To further investigate this phenomenon in Section 5.3, we conducted:
>
> 1. Two case studies: comparing Structured, Timelined, and Agent-based baselines; and
>
> 2. Ablation experiments: using varying CGM sequence lengths (30-day, 7-day, and 1-day)
> Results showed Timelined and Agent baselines, and a shorter context length of Structured baseline could mitigate this phenomenon to some extent, but the model still struggles to execute reliable programmatic calculations. Accuracy rises from 45.5% (30-day) to 39.0% (7-day) and 64.7% (1-day). The Timelined baseline achieved a higher accuracy of 56.8%,  Agent baseline achieved 77.3% accuracy (Appendix E, Table 3)
>
> We additionally discuss hypotheses regarding the root causes of this behaviour and relate our findings to prior studies in this section.
>
> (3)	Expanded Baselines Evaluation
>
> To provide comprehensive baselines, we have now compared GPT-5 across three prompting strategies (Section 5):
>
> 1. Structured prompting
>
> Presents the CGM session in a modality-separated format, where glucose readings and all event types are listed in distinct.
>
> 2. Timelined prompting
>
> Provides the same information in a unified chronological sequence, placing glucose values alongside insulin, meal, and exercise events under each timestamp (e.g., W1D1 10:20, 142.3, morning_snack 10.1g).
>
> 3. Agent-based prompting
>
> Uses the GPT agent framework with Python code-interpreter tools, enabling explicit numerical computation, data filtering, and programmatic reasoning during inference.
> Results showed Timelined could reduce the Temporal Misalignment error, and the agent baseline could effectively reduce both Reluctance to Calculate and Temporal Misalignment error.
>
> We appreciate the suggestion to compare against additional strong baseline models. We evaluated Qwen/Qwen3-30B-A3B-Thinking-2507, but due to our ~200k token long-context input, the model repeatedly entered prolonged reasoning loops and failed to produce final answers. And its “thinking” budget is not user-controllable. We aim to run more experiments on Sonnet during the rebuttal period.

---

### Official Review · Reviewer_ZaxN · 2025-11-02

**Soundness:** 3
**Presentation:** 3
**Contribution:** 3
**Rating:** 4
**Confidence:** 5

**Summary:**

The paper introduces a time-series reasoning benchmark designed for wearable diabetes care applications. The benchmark simulates single-turn question-answering (QA) interactions between a patient and an intelligent healthcare assistant, grounded in long-term continuous monitoring data. Each task instance includes (i) physiological time-series data, (ii) contextual information such as insulin delivery records, patient profile, and activity logs, (iii) a natural language question about the monitoring data, and (iv) a reasoning instruction specifying the evidence-gathering process. The model (based on an LLM) is expected to output a precise answer that may be numerical, categorical, or temporal in nature. The benchmark comprises 150 question templates spanning key medical time-series reasoning tasks, including process mining, anomaly detection, and prediction. To safely evaluate model robustness and system behavior, the authors generate an automatic insulin delivery systems monitoring dataset using a closed-loop in-silico testbed with 20 virtual Type 1 Diabetes patients, allowing for controlled simulation of device malfunctions and cyberattack scenarios without risk to real patients.

**Strengths:**

The paper is well-written, well-motivated, timely and the task is designed very well. I really like the design of the benchmark based on 3 key reasoning abilities (description, memory, and pattern-level), the break down of the reasoning process into atomic reasoning units (e.g, quantitative calculation, etc.), and the subsequent analysis of failure modes of LLM-based models.

**Weaknesses:**

- Missing details and reproducibility: I believe that the authors can significantly improve the reproducibility of their paper by including important details. I listed some questions in the following section.
- Baselines: The paper lacks baselines, which makes it hard to judge how good the proposed LLM-based model actually is. This is especially true for regression-based tasks, given that sMAPE / MAE are unbounded metrics.
- Reluctance to calculate: This is a well-known results in LLMs, which struggle with numerical calculations. I would recommend providing citations to prior work which has shown similar results.
- I would recommend adding descriptive captions to the figures and tables, which capture the story that they are communicating. Also, the current captions do not fully describe the different things shown in the tables, for example, it is unclear what $N$ (valid number of results) means.

### Minor
- Missing citations to prior work on time series question answering: There are a few benchmarks which have formalized the task of time series reasoning and question answering [1], which the literature work could be built on.
- The paper would benefit from proof-reading to correct minor grammatical mistakes and typos, for e.g., Srivastava et al. (2023); ?.

### References
1. Cai, Yifu, et al. "Timeseriesexam: A time series understanding exam." arXiv preprint arXiv:2410.14752 (2024).

**Questions:**

> We design 150 question templates covering core medical time-series tasks—process mining, anomaly detection, and prediction
- It is unclear how the question a generated from such a template. Can you provide an example of a template for each kind of time series task>

> In addition, each question is paired with a reasoning rationale that articulates the step-by-step logic behind the answer derivation
- How are these reasoning rationales generated? Are they generated automatically?

- What is the model that is used to derive results for Table 1? I would recommend adding a description of the LLM-based system, along with its hyper-parameters, and all other details necessary to reproduce the results.

---

> ### Author Response · Authors · 2025-11-27
>
> We thank the reviewer for the constructive and insightful feedback. We appreciate the recognition of HealthLoopQA’s novelty, systematic framework, realism, and alignment with the complexity of medical time-series data interpretation. In response to the identified weaknesses and suggestions, we have made the following revisions that improve clarity, reproducibility, and the breadth of empirical evaluation.
>
> 1.	Added the details of experiments and data generation
>
> To ensure reproducibility of the benchmark:
>
> •	We expanded Section 4.3 Dataset Collection with step-by-step descriptions of (1) AID monitoring data simulation, (2) question-template instantiation, and (3) answer extraction.
>
> •	Appendix H now includes illustrative examples of templates and generated questions for each task category.
>
> •	We clearly specified LLM system configurations, prompts, hyperparameters, and test-run details. The full prompts and implementation details are added in **Appendix F and Appendix G**.
>
> 2.	Expanded baseline evaluation
>
> To provide comprehensive baselines, we have now compared GPT-5 across three prompting strategies (Section 5):
>
> 1.	Structured prompting
>
> Presents the CGM session in a modality-separated format, where glucose readings and all event types are listed in distinct.
>
> 2.	Timelined prompting
>
> Provides the same information in a unified chronological sequence, placing glucose values alongside insulin, meal, and exercise events under each timestamp (e.g., W1D1 10:20, 142.3, morning_snack 10.1g).
>
> 3.	Agent-based prompting
>
> Uses the GPT agent framework with Python code-interpreter tools, enabling explicit numerical computation, data filtering, and programmatic reasoning during inference.
>
> Results showed Timelined could reduce the Temporal Misalignment error, and and agent baseline could effectively reduce both Reluctance to Calculate and Temporal Misalignment error.
> Weappreciate the suggestion to compare against additional strong baseline models. We evaluated Qwen/Qwen3-30B-A3B-Thinking-2507, but due to our ~200k token long-context input, the model repeatedly entered prolonged reasoning loops and failed to produce final answers. And its “thinking” budget is not user-controllable. We aim to run more experiments on Sonnet during the rebuttal period.
>
> 3. Expanded discussion of “In-Context Laziness”
>
> To further investigate this phenomenon in Section 5.3, we conducted:
>
> 1. Two case studies:  comparing Structured, Timelined, and Agent-based baselines; and
>
> 2. Ablation experiments: using varying CGM sequence lengths (30-day, 7-day, and 1-day)
> Results showed Timelined and Agent baselines, and a shorter context length of Structured baseline could mitigate this phenomenon to some extent, but the model still struggles to execute reliable programmatic calculations. Accuracy rises from 45.5% (30-day) to 39.0% (7-day) and 64.7% (1-day). The Timelined baseline achieved a higher accuracy of 56.8%,  Agent baseline achieved 77.3% accuracy (Appendix E, Table 3)
>
> We additionally discuss hypotheses regarding the root causes of this behaviour and relate our findings to prior studies in this section, situating our findings within existing literature on LLM reluctance to perform precise numerical computation.
>
> 4. Improved figure/table clarity
>
> We revised figure and table captions to be more descriptive (Figures 1–2, Table 2), clarifying variables and design choices.
>
> 5. Fixed citations and added missing related work
>
> We incorporated the reviewer’s suggestion to discuss TimeSeriesExam in Section 3. We proofread the paper and fixed citation format problems, including the mentioned one (Srivastava et al. (2023); ?).

---

> ### Author Response · Authors · 2025-11-27
> **Responses to Reviewer Questions**
>
> **Q1**: It is unclear how the question a generated from such a template. Can you provide an example of a template for each kind of time series task?
>
> A QA template comprises a natural language question, a dataset-agnostic answer extraction module, a context extraction module, and other metadata used in performance evaluation (e.g., answer instruction, metric). With a dataset simulated by the AID testbed, the precise answer and context (e.g., meal, exercise, and therapy events) are automatically generated for each question template.  For example, for question “What is my average blood glucose value”, we first define simulation scenarios (i.e., the range of meal’s time and carbs, exercise type, time, and intensity, and faults injection time and type), then based on such scenario generate 30 days uniformly sampled AID system monitoring data (e.g., CGM and insulin delivery). After getting the simulation data, then run the answer extraction module (in this case a simple average of the BG values) then pair it with question, monitoring data, context, and instruction to form a question from the template.
>
> We provided examples of a template for each kind of time series task in Appendix H.
>
>
> **Q2**: How are these reasoning rationales generated? Are they generated automatically?
>
> These reasoning rationales were carefully curated in collaboration with time series, security, and diabetes experts, following CGM data analysis standards and diabetes management guidelines, and cross-verified for accuracy.  While these reasoning rational can be programmatically run with any simulation scenarios, allowing flexible question data generation, the programmatic rule is defined manually by the above experts.
>
>
> **Q3**: What is the model that is used to derive results for Table 1? I would recommend adding a description of the LLM-based system, along with its hyper-parameters, and all other details necessary to reproduce the results.
>
> The original Table 1 results were obtained using GPT-5. Full model configurations, prompts, hyperparameters, and evaluation procedures are now provided in Section 5 and Appendices F and G.

---

### Meta-Review · Area_Chair_1Fnh · 2026-01-06

**Summary:**

Reviewers broadly recognized the need for AI for diabetes monitoring an praised the paper for the design, insights and the timeliness. Concerns remained about reproducibility, weaknesses, missing baselines and presentation quality along with generation processes. The testbed itself is quite impressive with realistic faults and the in-context laziness problem is quite interesting. However, there remain several concerns with baselines, reproducibility, presentation and the sim-2real situation.

**Reviewer Concerns:**

Some of the concerns with the presentation details are addressed in teh rebuttal (although I am not personally sure that just adding more and more appendix helps the problem).  Some concerns about presentation and execution also have been addressed reasonably well in the rebuttal. The question of missing comorbidities and behavioral variability is a harder one to address. And one cannot escape the single benchmark evaluation problem due to the nature of the paper.

Over all, a true borderline paper.

I had to FULLY read the paper myself. I tend to agree that this paper needs a major revision before it can be accepted.

**Reviewer Scores:**

I want to believe that the couple of reviewers who gave a 4 could go to a 5. I just do not see a very enthusiastic champion for the paper.

---

### Decision · Program_Chairs · 2026-01-26

Reject